# Quantum absorption refrigerator with trapped ions

Gleb Maslennikov[1], Shiqian Ding [1,3], Roland Hablützel[1], Jaren Gan[1], Alexandre Roulet [1], Stefan Nimmrichter[1], Jibo Dai[1], Valerio Scarani[1,2] & Dzmitry Matsukevich[1,2]

In recent years substantial efforts have been expended in extending thermodynamics to single quantum systems. Quantum effects have emerged as a resource that can improve the performance of heat machines. However in the fully quantum regime their implementation still remains a challenge. Here, we report an experimental realization of a quantum absorption refrigerator in a system of three trapped ions, with three of its normal modes of motion coupled by a trilinear Hamiltonian such that heat transfer between two modes refrigerates the third. We investigate the dynamics and steady-state properties of the refrigerator and compare its cooling capability when only thermal states are involved to the case when squeezing is employed as a quantum resource. We also study the performance of such a refrigerator in the single shot regime made possible by coherence and demonstrate cooling below both the steady-state energy and a benchmark set by classical thermodynamics.

[1] Centre for Quantum Technologies, National University of Singapore, 3 Science Dr 2, Singapore 117543, Singapore. [2] Department of Physics, National University of Singapore, 2 Science Dr 3, Singapore 117551, Singapore. [3] Present address: JILA, National Institute of Standards and Technology and University of Colorado, and Department of Physics, University of Colorado, Boulder, CO 80309, USA. These authors contributed equally: Gleb Maslennikov, Shiqian Ding. Correspondence and requests for materials should be addressed to D.M. (email: phymd@nus.edu.sg)

Thermodynamics is one of the oldest and best-established branches of physics that sets boundaries to what can be achieved in macroscopic systems. It also guided Planck's and Einstein's first steps into quantum mechanics. Decades later, it was realized that large quantum devices, such as masers or lasers, can be treated with the thermodynamic formalism[1,2]. Rapid progress in the experimental control of small quantum systems revives interest in the merging of thermodynamics with quantum mechanics[3,4] and poses fundamental questions: What is the smallest heat machine one can build[5]? Can quantum effects improve the performance of a heat engine, and if so, can we use quantum correlations as a fuel[6–9]?

Remarkable progress has been made recently in the miniaturization of heat machines[10] all the way to the single Brownian particle[11,12] as well as to a single atom[13]. While a lot of work in this field is focused on heat engines, we consider here another standard example of a heat machine: the absorption refrigerator. An absorption refrigerator consists of three parts: cold, hot, and work bodies (Fig. 1). Heat from the work body is used to cool down the cold one, while transferring heat to the hot body. The first such device was invented in 1850 by the Carré brothers[14] and was one of the first practical refrigerators used in the industry. In its improved design[15] it remains a popular choice of refrigeration devices[16]. Absorption refrigeration in the quantum regime has been the object of numerous theoretical studies[5,8,17,18], alongside

with proposed implementations with superconducting qubits[19,20], quantum dots[21], trapped ions[22], or optomechanical systems[23].

Here we implement a quantum absorption refrigerator utilizing three modes of motion of trapped Ytterbium ions, an axial and two radial ones, as the heat bodies (Fig. 1). The radial "zig-zag" mode represents the heat body that is in contact with a cold environment to be refrigerated, while the axial "zig-zag" mode of higher frequency represents the ambient (hot) temperature. The third radial "rocking" mode serves as the heat source that drives the refrigeration of the cold mode, replacing the work reservoir of a conventional refrigerator. We investigate the performance of the refrigerator in the quantum regime of low mean phonon numbers, including the case when the thermal state of the work mode is squeezed[18]. We also test whether there is an advantage when the refrigerator is operating in the single-shot cooling regime[24,25], which relies on coherent population oscillations that can occur among the coupled modes in the quantum system before a steady state is reached. While this behavior can also be explained qualitatively in a purely classical framework[26], the correct quantitative predictions for the observed low-excitation regime require a quantum description of the system.

## Results

**Principles of the refrigerator operation.** The interaction Hamiltonian in the system of three ions, induced by anharmonicity of the Coulomb repulsion between the ions, has the form[17,27] (Supplementary Note 1).

$$\hat{H} = \hbar\xi\left(\hat{a}_h^\dagger \hat{a}_w \hat{a}_c + \hat{a}_h \hat{a}_w^\dagger \hat{a}_c^\dagger\right), \tag{1}$$

where $\hat{a}_i$ $\left(\hat{a}_i^\dagger\right)$ are the annihilation (creation) operators for the corresponding harmonic oscillators labeled by $i = $ h, w, c, and $\xi = 9\omega_z^2\sqrt{\hbar/m\omega_h\omega_w\omega_c}/5z_0$ is the coupling rate. Here $z_0 = \left(5e^2/16\pi\epsilon_0 m\omega_z^2\right)^{1/3}$ is the equilibrium distance between the ions, $m$ is the ion mass, $e$ is the ion charge, $\epsilon_0$ is the vacuum permittivity, and $\omega_z$ is the single ion axial trap frequency. The Hamiltonian (1) is valid in the rotating wave approximation when the mode frequencies satisfy the resonance condition $\omega_h = \omega_w + \omega_c$. Away from this resonance condition, the energy exchange between the modes is suppressed[28,29].

The operation of the refrigerator in our system consists of three major steps. First, the Raman beams cool down all modes to the ground state of the trap and starting from there we selectively prepare the modes corresponding to refrigerator bodies in the desired states. Then, the resonant trilinear interaction is switched on for some time by tuning the mode frequencies to satisfy the resonance condition (Fig. 1) so that the modes start to exchange energy while the lasers are turned off. Finally, the mode frequencies are brought back to the initial values and the measurement of the resulting state in one of the modes is performed with the help of the Raman beams (Methods).

The refrigeration itself occurs during the second step and to see how it works[17], consider how the resonant interaction Hamiltonian (1) redistributes energy between the modes. The work mode (w) can remove one of its excess thermal phonons only by creating a hot mode (h) phonon and simultaneously annihilating a phonon in the cold mode (c). Hence the transfer of energy from (w) to (h) is always accompanied by energy transfer from (c) to (h). This can result in the cooling of the cold mode when the temperature of the work mode is higher than the hot mode and energy tends to flow from the former to the latter. At some temperatures the process is balanced by the flow of the energy in opposite direction, leading to an equilibrium.

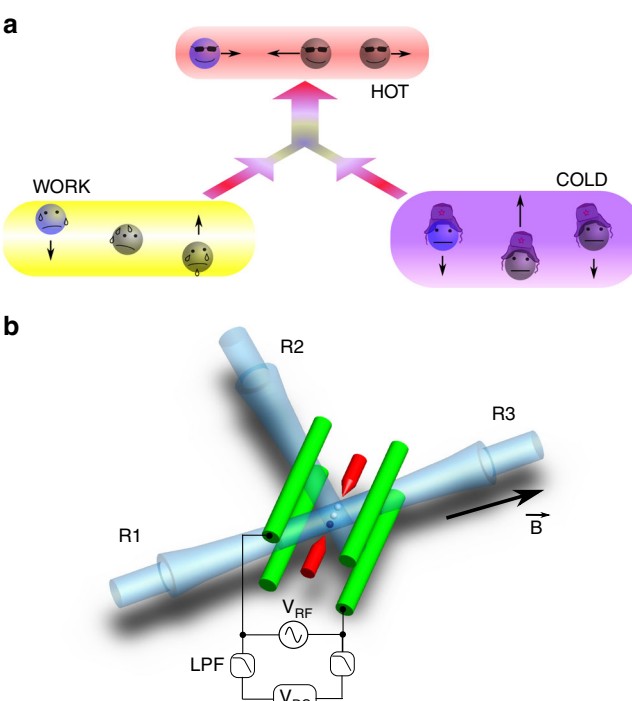

**Fig. 1** Experimental setup. **a** Direction of heat flow in the absorption refrigerator. Energy from the work body ("rocking" radial mode) is transferred to the hot body ("zig-zag" axial mode), which removes energy from the cold body ("zig-zag" radial mode). The black arrows label the motional eigenmodes utilized as heat bodies. **b** Schematic of the linear rf-Paul trap with three trapped $^{171}$Yb$^+$ ions. The Raman beams (R1, R2, and R3) are responsible for applying the optical dipole force required for state preparation, and for coupling the ions motional modes to the internal state during the motional state detection. Two (gray) ions are prepared in the $^2F_{7/2}$ "dark" state (see Methods). Radial confinement of the ions provided by radiofrequency (RF) potential can be fine tuned by adjusting the offset voltage applied to the diagonally opposite trap electrodes. The speed of this tuning is controlled by a pair of low-pass filters (LPFs)

For thermal states, at equilibrium the mean phonon numbers $\bar{n}_i^{(\mathrm{eq})}$ fulfill (Methods)

$$1 + \frac{1}{\bar{n}_\mathrm{h}^{(\mathrm{eq})}} = \left(1 + \frac{1}{\bar{n}_\mathrm{w}^{(\mathrm{eq})}}\right)\left(1 + \frac{1}{\bar{n}_\mathrm{c}^{(\mathrm{eq})}}\right). \qquad (2)$$

If the system is initially prepared away from equilibrium, under the interaction Hamiltonian (1) the initial mean phonon numbers $\left(\bar{n}_\mathrm{h}^{(\mathrm{in})}, \bar{n}_\mathrm{w}^{(\mathrm{in})}, \bar{n}_\mathrm{c}^{(\mathrm{in})}\right)$ of the heat bodies can only evolve as $\left(\bar{n}_\mathrm{h}^{(\mathrm{in})} - \epsilon_\mathrm{h}, \bar{n}_\mathrm{w}^{(\mathrm{in})} + \epsilon_\mathrm{w}, \bar{n}_\mathrm{c}^{(\mathrm{in})} + \epsilon_\mathrm{c}\right)$ such that $\epsilon_\mathrm{c} = \epsilon_\mathrm{w} = \epsilon_\mathrm{h}$. For a simple estimate of the achievable cooling in terms of mean phonon numbers, we assume an idealized scenario of operation where the states of the heat bodies remain thermal. Then the energy flow ceases when Eq. (2) is fulfilled, i.e. for

$$\epsilon = -\frac{1}{6}\left(a - \sqrt{a^2 - 12\left(n_\mathrm{c}^{(\mathrm{in})} n_\mathrm{w}^{(\mathrm{in})} - n_\mathrm{c}^{(\mathrm{in})} n_\mathrm{h}^{(\mathrm{in})} - n_\mathrm{h}^{(\mathrm{in})} n_\mathrm{w}^{(\mathrm{in})} - n_\mathrm{h}^{(\mathrm{in})}\right)}\right), \qquad (3)$$

where $a = 1 + 2\left(n_\mathrm{c}^{(\mathrm{in})} - n_\mathrm{h}^{(\mathrm{in})} + n_\mathrm{w}^{(\mathrm{in})}\right)$. Here, the system evolves towards correlated states that are not thermal: Eq. (3) will be used as a benchmark for comparison.

**Near-equilibrium operation**. To explore the parameter window of refrigeration and demonstrate the equilibrium performance of the refrigerator, we start with all modes prepared in thermal states (Methods) with $\bar{n}_\mathrm{h}^{(\mathrm{in})} \approx 0.6$ and various choices of $\bar{n}_\mathrm{w}^{(\mathrm{in})}$ and $\bar{n}_\mathrm{c}^{(\mathrm{in})}$. We then let the system evolve for long interaction times $\tau \gg \xi^{-1}$ (Supplementary Note 5) and average the measured mean phonon numbers of the hot mode to get an estimate of the asymptotic steady state value $\bar{n}_\mathrm{h}^{(\mathrm{ss})}$ (Fig. 2a–d). The cold mode is effectively cooled in those cases where $\epsilon_\mathrm{h} = \bar{n}_\mathrm{h}^{(\mathrm{in})} - \bar{n}_\mathrm{h}^{(\mathrm{ss})}$ is negative. Also, for each $\bar{n}_\mathrm{w}^{(\mathrm{in})}$ we extract from numerical fits to the data an equilibrium value for the mean phonon number $\bar{n}_\mathrm{c}^{(\mathrm{eq})}$ that corresponds to $\epsilon_\mathrm{h} = 0$. These points are plotted in Fig. 2e, demonstrating the validity of Eq. (2) in our setup. Refrigeration of the cold mode can be associated[16,17] to the three data points for which the temperatures $T_i = \hbar\omega_i[k_B\ln(1 + 1/n_i)]^{-1}$ satisfy the condition $T_\mathrm{c}^{(\mathrm{eq})} < T_\mathrm{h}^{(\mathrm{eq})} < T_\mathrm{w}^{(\mathrm{eq})}$ (here, $k_B$ is the Boltzmann constant).

**Steady state away from equilibrium**. We further notice in Fig. 2a–d that experimental points systematically disagree with Eq. (3) away from equilibrium. Indeed, numerical simulations of the evolution generated by the Hamiltonian (1) predict (Methods) that the mean phonon numbers approach the values of a non-thermal and correlated steady state in the long-time limit. That is, the system effectively equilibrates around the infinite-time average of its coherently evolving state, which we refer to as the asymptotic steady state, and the initial distribution of thermal energies does not recur even after interaction times much greater than $1/\xi$. This is related to the broad spectrum of incommensurate energy eigenvalues of the trilinear interaction Hamiltonian[26]: initially thermal states are diagonal in the three-mode Fock basis, but in the eigenbasis of the interaction Hamiltonian they exhibit non-diagonal elements with phases that quickly disperse and never fully rephase. As a result, the average energies of each mode undergo strong oscillations in a short transient time window smaller than $1/\xi$, after which they approach their long-time average values and only small residual fluctuations remain. The precise timing and the magnitude of the observed oscillations depends on the initial temperatures.

If the energy of a mode in the asymptotic steady state is lower than its initial energy, the mode is cooled down. For observation of cooling in the "cold mode", the initial energy of the work mode must satisfy[17,18] (see Methods section)

$$\bar{n}_\mathrm{w}^{(\mathrm{in})} > \bar{n}_\mathrm{h}^{(\mathrm{in})} \frac{1 + \bar{n}_\mathrm{c}^{(\mathrm{in})}}{\bar{n}_\mathrm{c}^{(\mathrm{in})} - \bar{n}_\mathrm{h}^{(\mathrm{in})}}. \qquad (4)$$

To investigate the cooling properties away from equilibrium, we now focus on the mean phonon number of the cold mode whose temporal evolution is shown in Fig. 3a–f. For $\bar{n}_\mathrm{h}^{(\mathrm{in})} = 0.66(4)$ and $\bar{n}_\mathrm{c}^{(\mathrm{in})} = 2.63(13)$, we observe a nett decrease of the cold mode mean phonon number in panels (a, b), equilibrium in (c), and an increase in (d–f). The data points are plotted relative to their long-time averages and show good agreement with theory. The nett difference of the long-time average values from the initial mean phonon numbers is shown in Fig. 3g. Again the predictions of quantum theory match the experiment, while the prediction of Eq. (3) disagrees with the data.

**Steady state for squeezed thermal states**. We next study the influence of quantum mechanical coherence on the cooling performance. There is still an ongoing intense debate on whether coherence degrades[30–32] or enhances[33] the performance of heat machines. In this work, coherence is added via squeezing the thermal state of the work mode. Several theoretical proposals appraise squeezing as a resource that allows heat machines to be more efficient[18,34,35] and to surpass even the Carnot limit[36]. We compare the cooling performance starting from a squeezed thermal state of the work mode[18] to the case where the mode is prepared in a thermal state with the same mean phonon number. In the experiment, an initially thermal state of the work mode at fixed $\bar{n}_\mathrm{w}^{(\mathrm{in})}$ is squeezed to several values of the squeezing parameter[37] $r$, which correspond to mean phonon numbers $\bar{n}_\mathrm{w}^{(\mathrm{in},sq)}(r) = \bar{n}_\mathrm{w}^{(\mathrm{in})} \cosh(2r) + \sinh^2(r)$. As $r$ increases in Fig. 3h–k, the system undergoes a transition from heating to cooling, which can be seen from the evolution of the cold mode. We plot the difference of final and initial mean phonon numbers in panel (l) for direct comparison to the previously discussed thermal case in (g). The experimental points agree with numerical predictions in panel (l). Furthermore, simulations show that the nett change in the mean phonon number is smaller, which implies that squeezing of the work mode decreases the cooling performance. For a given mean phonon number of the work mode, cooling is most effective when no squeezing is applied at all.

**Single-shot cooling**. We now focus on the single-shot cooling method[24,25]. Here the interaction is switched off at the right moment such that the evolution halts at a transient state with a lower mean phonon number $\bar{n}_\mathrm{c}$ than the long-time average. Conversely, one would achieve a higher mean phonon number in the heating regime. Both regimes can be seen in Fig. 3a–f, where the greatest deviation of $\bar{n}_\mathrm{c}$ from the initial $\bar{n}_\mathrm{c}^{(\mathrm{in})}$ is consistently reached at about 100 μs of interaction time, $\tau \approx (2\xi)^{-1}$. We plot in Fig. 4 the difference between the initial and this value for different work mode phonon numbers $\bar{n}_\mathrm{w}^{(\mathrm{in})}$. The first data point at $\bar{n}_\mathrm{w}^{(\mathrm{in})} = 1.3(1)$ has a vanishing difference since it corresponds to the system at thermal equilibrium according to (2). However, the difference increases with growing $\bar{n}_\mathrm{w}^{(\mathrm{in})}$, and consistently exceeds the difference from the long-time average value, demonstrating the advantage of coherence-assisted single-shot cooling[24]. Note that the cooling even exceeds the thermal equilibrium values set by the benchmark (3). However, to demonstrate the advantages of this method in practice, one would have to implement the trilinear interaction between the modes and couple each of the

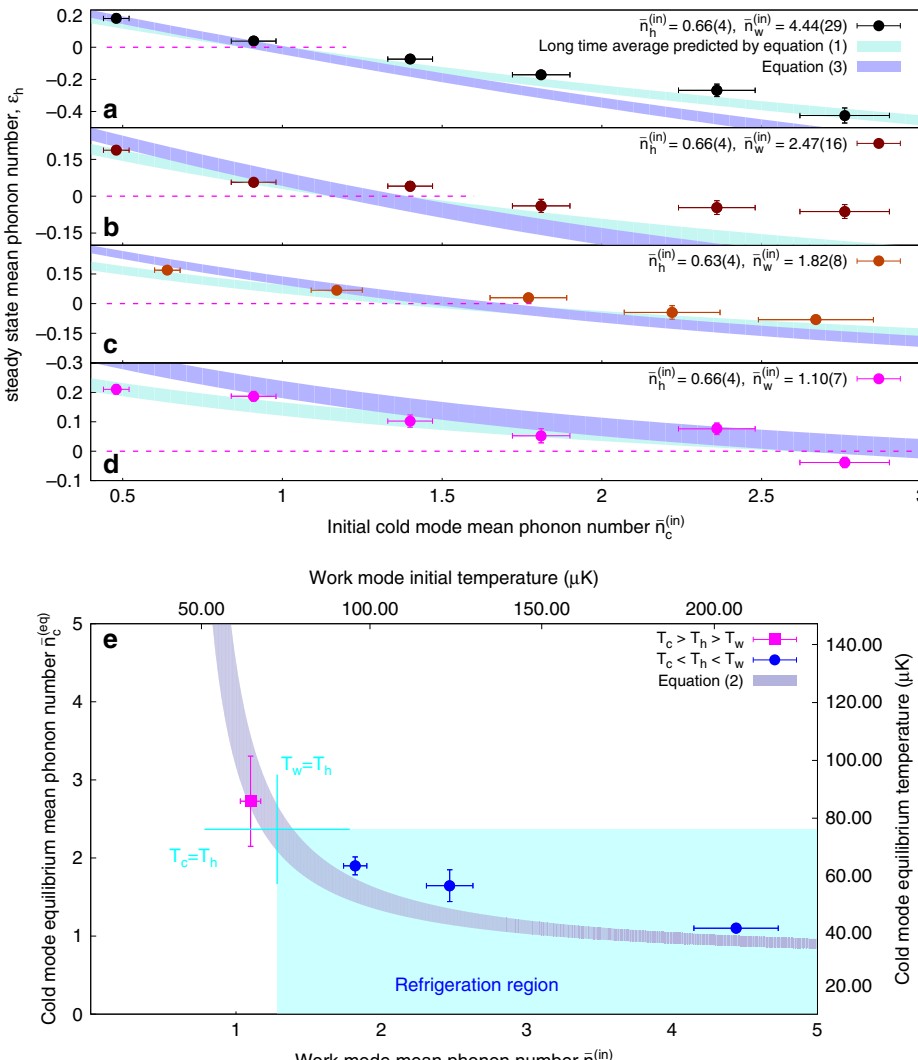

**Fig. 2** Absorption refrigeration demonstration. **a–d** The difference $\epsilon_h = \bar{n}_h^{(in)} - \bar{n}_h^{(ss)}$ of the initial hot mode phonon number and the asymptotic steady-state value plotted against the initial cold mode phonon number, $\bar{n}_c^{(in)}$ for different initial $\bar{n}_w^{(in)}$. The shaded curves are predictions of Eq. (3) (blue) and numerical simulations of (1) (turquoise), taking experimental uncertainty of initial state preparation into account. The numerical simulations of (1) agree well with the experiment. The equilibrium cold mode phonon number $\bar{n}_c^{(eq)}$, which corresponds to $\epsilon_h = 0$ (dashed line), is determined by fitting experimental data on **a–d** with the $\epsilon_h$ derived from numerical simulations of Eq. (1) using $\bar{n}_w^{(in)}$ and $\bar{n}_h^{(in)}$ as the fit parameters. Horizontal error bars in both panels are determined from the calibration of the initial state preparation and vertical error bars in **a–d** represent one standard error of the mean (SEM) (Supplementary Notes 3 and 4). **e** The values of $\bar{n}_c^{(eq)}$ are then plotted against experimentally prepared $\bar{n}_w^{(in)}$ and compared to the predictions of Eq. (2). The absorption refrigeration occurs at the region at which the cold mode temperature is the lowest (blue dots). For the magenta point, $T_c > T_w > T_h$. The vertical error bars are the fit errors of numerical simulations of data in **a–d** (Supplementary Note 4)

modes to its corresponding bath. The ability to cool more efficiently and on a shorter timescale is related to the transient coherence generated during the unitary time evolution under the trilinear Hamiltonian (1). Indeed, it is easily shown that an incoherent version of the trilinear interaction (Methods) precludes the device from cooling further than the steady-state energy[24,25].

In conclusion, we have demonstrated an implementation of an absorption refrigerator utilizing the harmonic modes of motion in a trapped-ion system. We have shown that the classical concept of the absorption refrigerator can be extended to the quantum domain. The experiment confirms our theoretical understanding of the refrigerator dynamics and its steady-state characteristics based on a coherent three-body interaction model. In particular, we could observe that, starting away from equilibrium, the system energies rapidly approach steady-state values, even in the absence

of environmental coupling. Simple arguments based on equilibrium thermodynamics do not predict these values, although they give the correct temperature requirements (4) for cooling. While it was shown that utilizing squeezed states allows the refrigerator to transition from a heating to cooling regime, hence demonstrating that squeezing could be used as a quantum fuel, our data also suggest a diminished performance of the refrigerator relative to thermal operation. This leads to the surprising implication that exploiting quantum resources does not necessarily enhance, but may even be detrimental to the performance of heat machines—an issue worth studying further.

## Methods
**Equilibrium and steady-state populations**. To gain insights on the operation of the absorption refrigerator we first consider an ideal adiabatic process

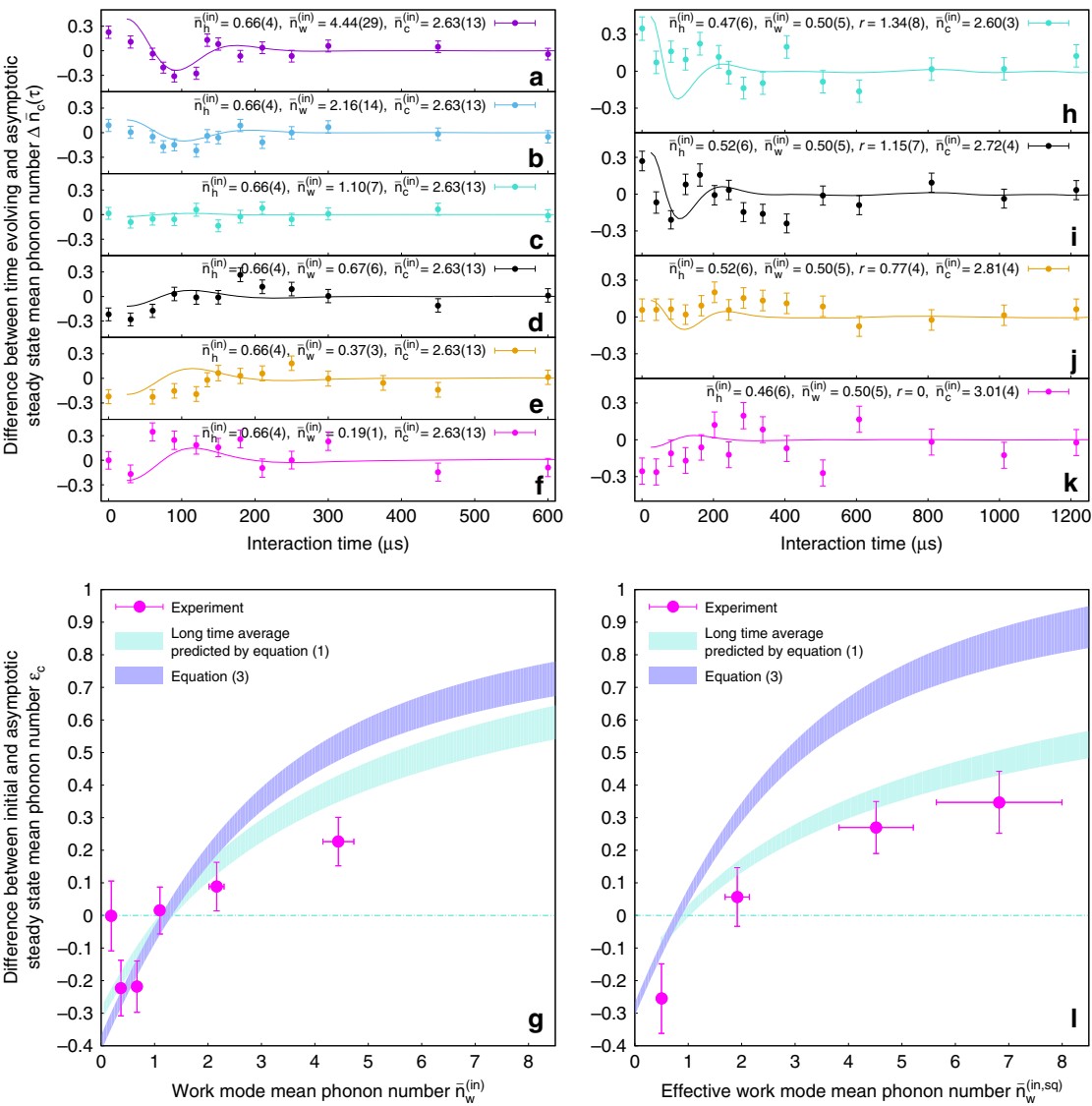

**Fig. 3** Non-equilibrium evolution of the cold mode with and without work mode squeezing. The difference $\Delta \bar{n}_c(\tau)$ between the measured time evolving $\bar{n}_c(\tau)$ and asymptotic steady-state values $\bar{n}_c^{(ss)}$ is shown as a function of $\bar{n}_w^{(in)}$ for the work mode initially prepared in a purely thermal state **a–f** and $\bar{n}_w^{(in,sq)}$ for squeezed thermal state **h–k**. The error bars are given by one SEM (Supplementary Note 4). The solid lines are numerical simulations of the state evolution using experimental initial conditions. The difference $\epsilon_c = \bar{n}_c^{(in)} - \bar{n}_c^{(ss)}$ of the initial cold mode phonon number and the asymptotic steady-state value plotted against $\bar{n}_w^{(in)}$ for thermal state **g** and $\bar{n}_w^{(in,sq)}$ for squeezed thermal **l** states of the work mode. The horizontal error bars are given by calibration of the initial state preparation and the vertical error bars represent one SEM (Supplementary Notes 3 and 4). The blue shaded curves show predictions of Eq. (3), while the turquoise shaded curves are numerical simulations of the state evolution under Hamiltonian (1). Both curves take into account the experimental uncertainty of initial state preparation. For **h–k**, initial mean phonon numbers of hot and cold modes were measured before each experimental run. Taking the average value of $\bar{n}_h^{(in)} = 0.49(3)$ to calculate the theoretical predictions of **l** mainly shifts the turquoise shaded line vertically compared to $\bar{n}_h^{(in)}$ of **g**

that satisfies[8,18]

$$\Delta \dot{S} = \frac{\dot{Q}_h}{T_h} + \frac{\dot{Q}_w}{T_w} + \frac{\dot{Q}_c}{T_c} = 0. \qquad (5)$$

Here $\dot{Q}_i = \hbar \omega_i \dot{n}_i$ is the energy per unit time flowing to the mode $i$ from its bath at temperature $T_i$. Using the canonical expression for the mean phonon number $\bar{n}_i$ for each mode, $1/T_i = \frac{k_B}{\hbar \omega_i} \ln(1 + 1/\bar{n}_i)$, and the constraint $\dot{n}_h = -\dot{n}_w = -\dot{n}_c$ implied by the Hamiltonian (1), Eq. (5) reduces to (2). The cooling condition (4) is obtained by noting that during cooling $\dot{n}_c < 0$ and $\Delta \dot{S} > 0$ (Second Law of Thermodynamics).

The quantum state $\rho = \rho_h \otimes \rho_w \otimes \rho_c$ corresponding to the equilibrium condition (2) is stationary as it commutes with the interaction Hamiltonian (1). Conversely, if the system is prepared out of equilibrium the trilinear interaction cannot drive it towards another equilibrium of this type. We nevertheless observe

(Fig. 3) that the unitary time evolution after long interaction time leads to an effective equilibration[38,39] of the mode energies around values corresponding to the infinite-time average of the system state, $\rho_\infty = \lim_{t\to\infty} \frac{1}{t} \int_0^t d\tau \rho(\tau)$. We refer to this as the asymptotic steady state, which can be computed by dephasing the initial ensemble in the eigenbasis of (1). The resulting density operator $\rho_\infty$ is not thermal and carries correlations between the three modes. The effective equilibration around this state is related to the specific nonlinear coupling Hamiltonian (1), which features a broad incommensurate energy spectrum and hence does not support a coherent rephasing of an initially uncorrelated thermal state under unitary evolution[26].

Note that by resorting to the unitary evolution of initially prepared thermal states, we are approximating the fast internal dynamics of an absorption refrigerator whose thermalization rate with the reservoirs associated to each mode is much slower. The effective equilibration we observe is an intrinsic feature that occurs on the fast timescale $1/\xi$ of the internal dynamics. On the other hand, high thermalization rates of the order of the coupling frequency $\xi$ would thwart the

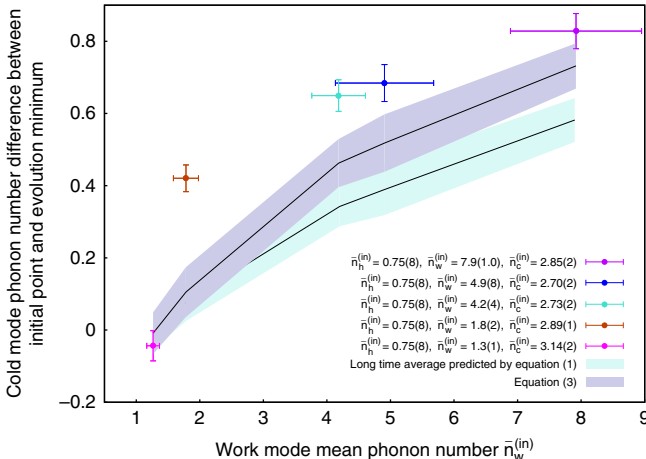

**Fig. 4** Absorption refrigerator operating in the single shot regime. The difference $\bar{n}_c^{(in)} - \bar{n}_c(\tau)$ between the measured initial phonon number and the mean phonon number at interaction time $\tau$ that gives the strongest cooling (colored points), is shown for several $\bar{n}_w^{(in)}$. The uncertainty in the $x$-axis is the error of the fit to the measured initial work population, while the uncertainty in the $y$-axis represents one standard error of the mean (Supplementary Note 4). The blue shaded region corresponds to values predicted by (3), while the turquoise shaded region is the long-time average predicted by numerical simulations. Both shaded regions take the experimental uncertainty of $\bar{n}_i^{(in)}$ into account

coherent dynamics required for single-shot cooling and keep the system close to the initial thermal state.

**Experimental setup.** The detailed description of our setup can be found elsewhere[40,41]. In brief, we trap three $^{171}$Yb$^+$ ions in a linear rf-Paul trap (Fig. 1a). The single ion trap frequencies are $(\omega_x, \omega_y, \omega_z) = 2\pi \times (1025.1, 937.7, 570)$ kHz for the data presented in Figs. 2 and 3a–g, and $(\omega_x, \omega_y, \omega_z) = 2\pi \times (764.9, 701.8, 425.3)$ kHz for Figs. 3h–l and 4. The radial frequencies are actively stabilized (drift < 200 Hz/hour) and can be fine tuned by DC offset voltages applied to two diagonally opposite trap electrodes, while the axial frequency is fixed and has negligible systematic drift. The normal modes chosen to represent the hot, work, and cold bodies are the axial zig-zag, the radial rocking, and the radial zig-zag mode (Fig. 1b), with frequencies $\omega_h = \sqrt{29/5}\omega_z$, $\omega_w = \sqrt{\omega_x^2 - \omega_z^2}$, and $\omega_c = \sqrt{\omega_x^2 - 12\omega_z^2/5}$, respectively (See Supplementary Fig. 1, Supplementary Table 2 and Supplementary Note 2 for details).

The modes are well isolated from the environment during an experimental shot. The heating rate of the modes is about 2 phonons/s. Fast fluctuations of the trap frequencies lead to additional decoherence between phonon Fock states; the smallest coherence times for superposition states of the form $(|0\rangle + |1\rangle)/\sqrt{2}$ exceed 8 ms. For the initial three-mode state $|1_h, 0_w, 0_c\rangle$, we could even observe the coherent energy exchange between the modes under the trilinear Hamiltonian evolution (1) for up to 60 ms (about 200 oscillations[41]) without significant reduction of the oscillation amplitude. We are thus confident that, for the interaction times used in the experiment, the modes experience an almost pure unitary evolution governed by the Hamiltonian (1).

A frequency-doubled, mode-locked Ti:Sapphire laser generates 250 mW with a central wavelength of 374 nm, pulse width of 3 ps, repetition rate of 76.2 MHz, and is used to achieve spin-motion coupling[42] and to apply the optical dipole force on the ion[43]. The beam is split into three paths R1, R2, and R3, as shown in Fig. 1. The R1–R2 pair addresses axial motion, whereas R2–R3 addresses radial motion. At all times, two of the three trapped ions are pumped into a dark metastable $^2F_{7/2}$ state and do not interact with the laser beams[40]. The remaining ion is always positioned at the edge of the ion chain to enable addressing of all the modes of motion. The positioning is accomplished by monitoring the fluorescence of the ions on an EM-CCD camera for ≈200 ms between each 100 experimental shots. If the ion jumps to the center, for example due to collisions with background gas, the RF signal sent to the trap is briefly interrupted for a few µs. This melts and recrystallizes the ion crystal, and is repeated until the bright ion is found at the edge of the chain. We use standard optical pumping to initialize the ion in the $|\downarrow\rangle \equiv \left|S_{1/2}, F = 0, m_F = 0\right\rangle$ state. The state $|\uparrow\rangle \equiv \left|S_{1/2}, F = 1, m_F = 0\right\rangle$ is detected by means of resonance fluorescence[44]. The optical dipole force is applied to the ion in the state $|a\rangle \equiv \left|S_{1/2}, F = 1, m_F = +1\right\rangle$.

**Experimental sequence.** The experiment starts with Doppler cooling of the ion chain (6 ms) followed by Sisyphus cooling[45] (15 ms) and Raman sideband cooling of all 9 modes (≈30–40 ms). The residual mean phonon numbers after each cooling stage are $\bar{n}_{Doppler} \approx 20 \rightarrow \bar{n}_{Sisyphus} \approx 1 \rightarrow \bar{n}_0 \leq 0.05$. The optical pumping pulse (5 µs) prepares the ions in the internal state $|\downarrow\rangle$. A microwave $\pi$-pulse (5.6 µs) then transfers the ion from $|\downarrow\rangle$ to $|a\rangle$, where the preparation of a thermal or squeezed thermal state occurs (≈5–15 ms, depending on the state), and a second microwave $\pi$-pulse brings the ions back to $|\downarrow\rangle$. Another optical pumping pulse is applied to remove the residual population in state $|a\rangle$ to $|\downarrow\rangle$. These operations are carried out at detunings $\Delta = \omega_a - \omega_b - \omega_c \approx -2\pi \times 80(-2\pi \times 40)$ kHz, for the high (low) single ion trap frequencies. For all cases, the detuning $\Delta \gg \xi$ so that the coupling between the modes is effectively switched off. The interaction can then be switched on for a time $\tau$ by bringing the detuning to $\Delta = 0$ and then switched off again. To perform motional state detection, we drive the red sideband on the $|\downarrow\rangle \rightarrow |\uparrow\rangle$ transition for about 60 µs, followed by 1 ms long state detection of $|\uparrow\rangle$.

Each experimental shot takes at most 70 ms, and the experiment is divided into groups of 100 single shots. Between subsequent groups, the experiment may be interrupted if additional adjustments are required. For instance, we regularly check whether the resonance condition $\Delta = 0$ still holds. For that we measure the avoided crossing between the modes[29] and, if necessary, adjust the DC offset voltages so that the frequency splitting between the eigenstates near resonance is minimized. If no intervention is needed, the experiment is continued until the desired amount of data for a given interaction time $\tau$ is collected. The typical number of experimental shots per point is on the order of $N = 7000$ in Figs. 2, 3, and can be as high as 20,000 for the data in Fig. 4. A single time evolution dataset plotted in the upper panels in Fig. 3 thus takes around 2 h of experimental time. Technical interventions, such as re-positioning of the bright ion to the edge of the chain, frequent checks of the resonance condition, and adjustment of the trap frequencies can triple the amount of experimental time. For the data presented in Fig. 4, we alternate experimental shots between $\tau = 0$ and $\tau \approx (2\xi)^{-1}$. This allows us to minimize the effect of systematic errors such as slow trap frequency drifts, which are inevitably acquired during the long data taking employed for Figs. 2, 3. In the following, we describe all stages of the experiment in detail.

**State preparation.** In order to prepare a thermal state, we transfer the bright ion into the state $|a\rangle$ and excite its motion with modulated optical dipole force. The force along the axial (radial) direction is applied by a running optical lattice formed by two linearly polarized beams R1 and R2 (R3 and R2) with orthogonal polarizations (Fig. 1a)[40,43]. The frequency difference between these two beams is set to match the frequency of the target mode while the phase of one of the beams is changed randomly every 100 µs step. The motional state of all three ions undergoes a random walk in phase space, which leads to a thermal state if the number of steps is large enough[46]. Typically we apply from 7 to 40 steps for state preparation. The final mean phonon number of the thermal state $\bar{n}$ after $N$ steps is

$$\bar{n} = \bar{n}_0 + N\bar{m}, \qquad (6)$$

where $\bar{n}_0$ is the mean phonon number after sideband cooling, and $\bar{m}$ is the mean phonon number of a coherent state after applying a single 100 µs step to the initial vacuum state (Supplementary Note 3).

The squeezed thermal state is generated by application of the squeezing operator $\hat{S}(z) = \exp\left((z^*\hat{a}^2 - z\hat{a}^{\dagger 2})/2\right)$ to a thermal state[37,47] where $z = re^{i\theta}$ and $r$ is squeezing parameter. Experimentally, the squeezing operation is realized by applying an optical dipole force produced by an optical lattice running at twice the mode frequency[40,43,48]. The squeezing parameter $r$ is linearly proportional to the duration of this step (see Supplementary Note 3).

**Energy exchange.** After preparing the motional modes we adjust the offset voltages applied to trap electrodes via third order RC low pass filters (LPF) with a 3 dB point at 11 kHz. This brings the modes to resonance ($\Delta = 0$) with a delay of 25 µs which is much smaller than $1/\xi$. The coupling rate is measured to be $\xi = 2\pi \times 2.64$ (5) kHz for data presented in Figs. 2 and 3a–g, and $\xi = 2\pi \times 1.89(4)$ kHz for Figs. 3h–l and 4. After interacting for time $\tau$, the offset voltages are reverted back to initial values and the motional modes are decoupled. The motional states are then mapped onto the internal state of the ion for state analysis[48,49].

**Motional state detection.** State detection of a mode of interest, after some interaction time $\tau$, can be done by measuring the probability $p_\uparrow(\tau)$ to find the detection ion in the "bright" internal state $|\uparrow\rangle$, after driving a red motional sideband between $|\downarrow\rangle$ and $|\uparrow\rangle$ with a pulse of fixed duration $t_{rsb}$. This probability is dependent on the population distribution $p(n, \tau)$, and has the form

$$p_\uparrow(\tau) = a + \upsilon \sum_{n=0}^{\infty} p(n, \tau) \frac{(1 - \cos(\sqrt{n}\Omega t_{rsb}))}{2} e^{-\gamma\sqrt{n}t_{rsb}}, \qquad (7)$$

with $\Omega$ the Rabi frequency of the red sideband, $a$ the background contribution to the state detection probability, and $\gamma$ is the decoherence rate between motional states. The factor $\upsilon$ is defined as the probability to detect an ion in the state $|\uparrow\rangle$ after

a $\pi$ pulse on a blue sideband transition, $|0,\downarrow\rangle \rightarrow |1,\uparrow\rangle$, where the first index corresponds to the motional Fock state. The values of $t_{\rm rsb} = 2\pi/(3\Omega)$ ($t_{\rm rsb} = \pi/(3\Omega)$) were chosen to maximize the sensitivity of $p_\uparrow(\tau)$ to the mean phonon number variations around 0.6 (2.7) for the measurements performed on the hot (cold) mode. For this choice of detection pulse time and detuning from resonance, the contribution of adjacent "spectator" modes to the detected signal $p_\uparrow(\tau)$ is negligible.

Typically, the population distribution is expected to have some analytic time-independent form. In this case the detection pulse duration $t_{\rm rsb}$ solely defines $p_\uparrow(\tau)$ and one can compute the inverse function $p_\uparrow^{-1}(\tau)$ that links the measured ion brightness $p_{\uparrow \exp}(\tau)$ to mean phonon number $\bar{n}(\tau)$. However in our experiment, during the interaction the states evolve away from their initial thermal (or squeezed thermal) population distribution. Then $p(n,\tau)$ is not known a-priori. We therefore compute for each given set of initial states $\left(\bar{n}_{\rm h}^{\rm (in)}, \bar{n}_{\rm w}^{\rm (in)}, \bar{n}_{\rm c}^{\rm (in)}\right)$ the expected spin-flip probability $p_{\uparrow \rm th}(\tau)$ and the mean phonon number $\bar{n}_{\rm th}(\tau)$ during the state evolution, by numerically solving (1). In order to obtain the best estimate for the experimental mean phonon numbers, we determine $\bar{n}_{\exp}(\tau)$ from the experimentally measured spin-flip probability $p_{\uparrow \exp}(\tau)$ using:

$$\bar{n}_{\exp}(\tau) \approx \bar{n}_{\rm th}(\tau) + \frac{\partial \bar{n}_{\rm th}(\tau)}{\partial p_{\uparrow \rm th}(\tau)}\left[p_{\uparrow \exp}(\tau) - p_{\uparrow \rm th}(\tau)\right]. \tag{8}$$

The partial derivative in (8) is approximated by

$$\frac{\partial \bar{n}_{\rm th}(\tau)}{\partial p_{\uparrow \rm th}(\tau)} \approx \frac{\bar{n}_{\rm th}\left(\tau; \bar{n}_i^{\rm (in)} + \delta\right) - \bar{n}_{\rm th}\left(\tau; \bar{n}_i^{\rm (in)} - \delta\right)}{p_{\uparrow \rm th}\left(\tau; \bar{n}_i^{\rm (in)} + \delta\right) - p_{\uparrow \rm th}\left(\tau; \bar{n}_i^{\rm (in)} - \delta\right)},$$

where $\bar{n}_i^{\rm (in)}$ is the initial population of the mode and $\delta$ is a small but finite deviation from $\bar{n}_i^{\rm (in)}$. For example, if the cold mode is the mode of interest, numerical simulations of $\bar{n}_{\rm c}(\tau)$ would be carried out for $\bar{n}_{\rm c}^{\rm (in)} \pm \delta$ with fixed $\bar{n}_{\rm h}^{\rm (in)}$ and $\bar{n}_{\rm w}^{\rm (in)}$. We have tested that the value of the partial derivative in Eq. (8) is robust to the choice of $\delta$ in Eq. (8), and to the changes of the phonon number distribution due to variations of the detuning $\Delta$ from resonance ($\partial \bar{n}_{\rm th}(\tau)/\partial p_{\uparrow \rm th}(\tau)$ changes less than 1% for $\Delta = 200$ Hz). Also, Eq. (7) does not include off-resonant contributions of the carrier transition which Rabi frequency depends on the population of the excited modes. However, we have estimated that this effect gives systematic errors of less than 1% to the reconstructed mean phonon numbers.

**Numerical simulation.** The interaction Hamiltonian (1) couples Fock states of the form

$$\{|n_{\rm h}, N - n_{\rm h}, M - n_{\rm h}\rangle : 0 \le n_{\rm h} \le \min(N, M)\} \tag{9}$$

with fixed integers $N$ and $M$. This basis spans a finite-dimensional Hilbert space. The evolution of the three-mode state is then computed by diagonalizing the Hamiltonian in each of the contributing subspaces, up to a cutoff for both $N$ and $M$. For all the simulations presented in this paper, the cutoff has been chosen to ignore terms in the initial density matrix smaller than $10^{-4}$. We also implemented an incoherent version of the interaction by integrating the master equation $\partial_t \rho = -\xi_{\rm in}[\hat{H}, [\hat{H}, \rho]]$, which describes an exponential decay of coherences in the eigenbasis of the Hamiltonian at the rate $2\xi_{\rm in}$. The fully dephased asymptotic state of this master equation predicts the infinite-time average phonon numbers of the coherently evolving state. However, the incoherent model does not reproduce the single-shot cooling behavior[26].

## Data availability

The data that support the findings of this study are available from the corresponding author upon request.

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

## Acknowledgements

We acknowledge discussions with Alex Kuzmich, Atac Imamoglu, and Mark Mitchison. This research is supported by the Singapore Ministry of Education through the Academic Research Fund Tier 2 (Grant No. MOE2016-T2-1-141) and Tier 3 (Grant No. MOE2012-T3-1-009); by the National Research Foundation, Prime Ministers Office, Singapore, through the Competitive Research Programme (Award No. NRF-CRP12-2013-03); and by both above-mentioned sources, under the Research Centres of Excellence programme.

## Author contributions

S.D. and D.M. conceived the experiment. S.D. setup the experiment and took the initial round of data. G.M. and R.H. improved the experimental setup and took the final data with the help of J.G., A.R., S.N., J.D., and V.S. provided theoretical support. All authors contributed to the data analysis and preparation of the manuscript.

## Additional information

**Competing interests:** The authors declare no competing interests.

