## [Peer Review File · Nature Communications]

Reviewers' comments:

Reviewer #1 (Remarks to the Author):

In their manuscript „Quantum absorption refrigerator with trapped ions“, Maslennikov et al. describe an experiment where three collective vibrational modes of an ion crystal are controllably tuned to a three-mode resonance, such that a sum frequency generation process can take place, where two phonons of a “work” and “cold” mode are annihilated and a phonon in a “hot” mode is created. The authors prepare the modes in thermal states with predefined mean phonon numbers, and measure resulting steady-state mean phonon numbers as well as the temporal evolution. In addition, the authors investigate the case where the “work” mode is prepared in thermal squeezed states, and investigate the effect on the system dynamics and steady-state. Furthermore, the authors demonstrate a “single-shot” cooling technique, where the time the system spends on the resonance is chosen to enforce maximum depletion of the cold mode.

Certainly, “quantum thermodynamics” is a timely subject, with a massive imbalance between theoretical and experimental work, and the idea of using a three-ion crystal with a nonlinear resonance as an experimental platform for studying refrigeration in the quantum domain is compelling. However, careful assessment of this work leads to concerns about the underlying concepts, the evaluation of the data, the question whether the data supports the claims made by the authors, and the interpretation. I list my main concerns:

-Unitary refrigeration? This work deals with closed system with three degrees of freedom, far away from the thermodynamic limit. Switching on the interaction Hamiltonian Eq. 1 causes a unitary evolution, which cannot change the entropy of the entire system. Therefore, can a “steady state” be achieved at all, where all modes are in thermal states? Would the process not be reversed at some point without additional non-unitary processes, and the “cold” mode would assume its initial state? Why does something like a steady-state occur here? This issue could probably be addressed by an analysis showing that the Poincare recurrence time by far exceeds any other timescale of the system.

The authors vaguely address this issue in the paragraph at l74 and in the methods section l140, clarification is however not achieved.

Furthermore, the authors seem to use “steady-state” and “equilibrium” interchangeably, this should also be clarified.

Concerning the “single-shot cooling” method, it is hard to judge how this can be linked to actual refrigeration, as this is a finite-time unitary operation. If we imagine a persistent heat load on the “cold” mode, would it be possible to apply such “single shot cooling” repeatedly to maintain the “cold” mode at a low temperature?

I would like to illustrate this with an example: If we prepare a vibrational mode of an ion crystal in a thermal state, drive a rapid adiabatic passage state transfer on the red motional sideband, and subsequently monitor and analyze Rabi oscillation on the blue motional sideband, we would observe a decreased “temperature”. However, it is obvious that the rsb state transfer is a unitary operation rather than cooling.

-Data evaluation: I am wondering about the small vertical error bars in Fig 2 a-d. E.g., for panel d) at $\bar{n}_c^{(in)} = 0.5$, the horizontal error is about ± 0.036 , while it is only 0.013 for the hot mode mean phonon number *difference*, which results from *two* measurements. This led me to look at the paragraph “Motional state detection” in the Methods section. Here, the authors state that rather than recording Rabi oscillations on one or more motional sidebands and extracting the required mean phonon number, they one probe the red sideband of the respective mode at a single fixed time t_{rsb} at a net Rabi frequency Ω . Both parameters are not indicated. It is also not explained if t_{rsb} is chosen differently for different modes or particular measurements. The authors link some observed probability to measure the ion in “spin up” after driving the sideband for t_{rsb} via Eq. 6. As the phonon distribution $p(n)$ is not given, the authors precompute some expected distribution, leading to an expected mean phonon number

\bar{n}_{th} , and use the measured probability $p_{\uparrow exp}$ to obtain a corrected “experimental result”.

I have concerns about the validity of this approach. If the measurement would contain no information about the desired quantity, we would have $\frac{\partial \bar{n}_{th}}{\partial p_{\uparrow th}}=0$, and the Eq. 7 would merely return the precomputed value. In other words, the results are biased towards an expected outcome. It is also unclear how the authors obtain the shown confidence intervals for measured mean phonon numbers. Furthermore, the coupling to spectator modes while driving the red sideband is not taken into account in Eq. 6. While the beams R1, R2, R3 are shown in Fig. 1, the authors do not explain which beams are used for coupling to which mode. Using R2 and R3 for driving the rsb of the “cold” mode on one of the outer ions, one would expect parasitic coupling to the “work mode”, .i.e. the resulting mean phonon number on the “cold” mode would depend on the state of the “work” mode. These issues raise concerns about the validity of all measurement data presented in this work.

-Quantum coherence: The authors claim to study the “influence of quantum mechanical coherence” by preparing the “work” mode in thermal squeezed states. The motivation, results and interpretation are rather unclear to me. It seems that the squeezing operation merely increases the mean phonon number of the “work” mode, which leads to increased depletion of the “cold” mode. The question whether the cooling efficiency is increased or not as compared to a thermal state of the same effective mean phonon number cannot be answered on the basis of the data presented in Fig. 3, in contrast to the claim in 195-197. It also remains unclear if the discrepancy in the simulations (green areas in Fig 3. G & I) is actually due to quantum coherence in the work mode, or merely due to different higher momenta in the phonon statistics. The authors do not explain why they use squeezing rather than displacement, which is easier to achieve and also serves to increase the work mode mean phonon number. I suspect that squeezing was chosen because it sounds more “quantum”.

-Data quality & claims: The quality of the data seems to be insufficient to support some claims made by the authors. For Fig. 2 a-d, the authors state “The numerical simulation agree well with the experiment.” This is certainly not the case for panel b, which is not discussed. The data in Fig. 2e results from a very gross evaluation method of connecting two points around the vertical “equilibrium” lines in panels a-d. From visual comparison with panels a-d, the vertical error bars in e seem to be substantially underestimated.

In Fig. 3 panels a-f,h-k, the data exhibits only a very vague tendency to have something to do with the simulations, which supports doubts about the data evaluation method. The data presented in panels g & I does not allow to draw any conclusion on different performances of thermal and squeezed thermal states. Why is there no data point in panel g for $\bar{n}_w(in) > 5$, although there is one in Fig. 4? For thermal states of the “work” mode, only one data points supports the general observation of cooling with significance. This cooling amounts only to less than 10% of the initial thermal energy of the “cold” mode.

In conclusion, this could be an interesting work if it would be more thorough on analyzing and discussing the validity and applicability of the underlying concepts, and if the data acquisition and evaluation would be more trustworthy. In the present state, this manuscript is not suitable for publication in Nature Communications.

I list a number of further issues below:

Fig. 2e: The authors indicate a “no cooling” region. Would the region above $T_c=T_h$ not also be a “no cooling” region?

p7, 178: The statement leading to Eq. (3) seems to be logically disconnected from the text before and after it.

Fig 3. I): The blue shaded area is supposed to pertain to Eq. 2, which however holds only for thermal states. The discrepancy with respect to panel g is not explained.

Fig 3, caption: The statement “A slight difference...” should be put into some context.

p11, l118: the comparison with an actual refrigerator would be suitable for an entertaining press release or feature article, but not for a scientific publication. The authors compare a *part* of a classical refrigerator to their system, where the three ions comprise the coolant, the material to be cooled down, and part of the machinery. Here, one could debate if the trap could be seen as the 'compressor', as it is required to tune the crystal to the parametric resonance. Then, the quantitative comparison in terms of mass would lead to a very different result.

p11, l125: "...hence demonstrating that squeezing could be used as a quantum fuel" contradicts somewhat with the statement on p9, l96 "...which implies that squeezing of the work mode decreases the cooling performance".

Eq. 4: It is not clear to me if this equation (along with the second law below) holds for the system, which is far away from the thermodynamic limit. Extracting/adding heat (\dot{Q}) from/to a finite-sized system will substantially change its temperature. A constant temperature at finite heat extraction/addition is only maintained for an infinitely large system (a so-called reservoir).

l136: here, Boltzmann statistics rather than Bose Einstein statistics apply, the formula however is correct.

l140: It is unclear what "The corresponding quantum state" refers to.

l145: "...obtained by dephasing the initial ensemble.." It is unclear if "dephasing" refers only to a computation, and if yes, if some actual dephasing takes place on relevant timescales, how it is caused, and what its implications would be. Furthermore, the implication of the correlations is unclear. The entire paragraph is rather confusing.

l148: also unclear: the timescale for "the fast internal refrigerator dynamics" should be ξ . The solid lines in Fig. 3 indicate that what the authors seem to call "thermalization" happens on the same timescale, with a factor ~ 2 difference. The statement implies a timescale separation, and it implies some sort of control parameter.

l160: what is the "coherence time of a single phonon"?

l167: "is always positioned at the edge of the ion chain" how is this achieved?

p14 "state preparation": it should be noted which beam pair is controlling which mode, and the authors should estimate the impact of off-resonant excitation of spectator modes during the preparation of one particular mode. The Lamb-Dicke parameters and the resulting \bar{m} values for each mode should be indicated.

l197: If the low-pass filters are mentioned, also the order and the cutoff should be specified.

Eq. 6: The choice of η as a contrast is unfortunate, as this commonly denotes a Lamb-Dicke parameter, which is however absorbed into Ω . It should be discussed why spectator modes are not included here. In Eq. S1, the authors include an additional contrast decay, why is this not included in Eq. 6. The number of experimental shots per data point is nowhere mentioned.

Reviewer #2 (Remarks to the Author):

Small thermal machines consisting only of a few degrees of freedom have recently attracted considerable attention, and this manuscript presents an experimental study in this context. The evolution of three resonantly coupled vibrational modes is such that one of these modes are cooled due to the flow of energy between the others, analogous to an absorption refrigerator.

The authors test whether the performance of this device is better when one of the modes is initially in a squeezed thermal state rather than being thermal. They also compare the performance for the long-term evolution of these modes, with a specifically selected short-time evolution. They find that in the long-time regime the coherence in the squeezed state can act as a fuel and induce cooling, but that the performance of the device can be hampered rather than helped by the squeezing. They also find that for the specifically selected short-time evolution, the performance of the device can exceed that of the long-term operation, and be enhanced by the squeezing.

As far as I am aware there is no previous experimental studies of this particular setup in the context of quantum thermal machines, and there are generally still relatively few experiments that

probe thermal machines at the quantum level. I would unfortunately not expect that the detailed findings as such would have a significant impact on the field, but the general approach could potentially inspire future experiments and theoretical studies.

Unfortunately, I am somewhat concerned about how the results are presented in relation to the existing notions of thermal machines, but I believe that the authors potentially could improve the clarity of the presentation, as well as the potential for impact, by suitable modifications and additions. Below I try to convey the nature of these concerns, and provide some suggestions.

The absorption refrigerator and many other thermal devices are, both in the standard classical case and in several quantum generalizations, often analyzed in setups where the device is coupled to infinite ideal heat reservoirs that can maintain constant temperatures in spite of indefinite flows of energy through the device. This enables an approach where one analyzes the performance in terms of steady state flows that acts like attractors to the dynamics of the system. In the recent literature on quantum thermal machines, this steady state performance is often compared with the "single-shot" performance, where we select a specific optimal time where the operation of the device is terminated. It appears that the authors aim to study an analogous scenario in their experimental setup.

As far as I understand, the authors do in their main analysis regard the three modes as a closed system that evolves according to the Shrodinger evolution governed by the Hamiltonian (1). In particular I get the impression that they do not add any effects of couplings to an environment (for example via master equations). I would say that such a setting of isolated systems is quite different from a scenario where the systems are attached to ideal reservoirs. In particular, one cannot for isolated systems have steady states in the sense of attractors of the dynamics, and it is in this case also problematic to discuss the evolution of observables in terms of "long time limits". (If the authors indeed wish to claim that some observables display a non-trivial limit in this system, I would urge them to prove this claim, or to cite some source where this is proved.) Unfortunately, the manuscript does repeatedly describe the closed system evolution in terms of "steady states" and "long time limits". Moreover, the whole presentation may give the unfortunate impression that it almost equates the reservoir scenario with the closed system scenario, and I am afraid that this may cause considerable confusion.

For this type of closed system I would expect that the observables, instead of evolving to a proper limit, would evolve in a quasi-periodic manner. However, it may very well be the case that the relevant observables may stay close to an "equilibrium value" for long stretches of time, although deviating from it occasionally. I get the impression that the authors are aware of this, due to the paragraph in the Methods-section on page 12 where they cite [28] and [29], where these notions are discussed.

Thus, an alternative way to phrase the results of this study would be to say that it compares the performance of this isolated thermal machine, for the case when the evolution time is selected specifically to optimize the performance, with the case when we pick "generic" long evolution times, where the relevant observable is close to its equilibrium value, in the sense of closed system equilibration. I would thus suggest that the authors consider the possibility to, instead of phrasing their study as a direct quantum implementation of the standard notion of absorption refrigerators, phrase it as an analogous construction, but in a closed system scenario, and explicitly discuss and highlight both differences and similarities between the two. (I would also avoid to use the terms "steady state" and "long term limits" in the closed system scenario.) I also think it could be fruitful to explicitly discuss (and potentially investigate a bit further) to what extent one can understand the behavior of this device in terms of closed system equilibration. I might add that this topic has witnessed a rather vigorous development, and a clear connection could potentially boost a wider interest for this study.

All the above comments concerning "steady states" and "long term limits" relate to the closed system scenario.

However, another possibility is that there actually is some additional irreversible influence from the environment. For example, in addition to the Hamiltonian evolution there could be some degree of decoherence with respect to the eigenbasis of the global Hamiltonian (for example as described by the master equation on page 17, although one could of course combine the Hamiltonian evolution with a dissipator of this type). This would yield an irreversible approach to the decohered state in the energy eigenbasis, and a well defined limit state. (Limit states may also exist for other open system effects that do yield decohere to the energy eigenbasis.)

If the experimental conditions are such that there is a significant degree of decoherence, or not, is beyond me to judge. Unfortunately, the only comment that I can find in the manuscript that touches this issue is on page 13 in the Methods-section, where it is stated that the coherence time of a single phonon is much larger than the time required to achieve the "steady state". The question whether the evolution in this experiment can be regarded as purely Hamiltonian to a good approximation, or if there are some significant open system effects, appears to me to be rather fundamental for this study, and I think that it deserves more attention and should be discussed more thoroughly. Moreover, if it indeed would be obvious that the evolution is Hamiltonian, and described by (1), then one might wonder why the experiments were done in the first place, as it then seems like one equally well might have resorted to purely numerical investigations. I cannot judge the feasibility, but maybe the authors could consider the possibility to complement their study with some more direct probing of the quality of the Hamiltonian description for this system? Moreover, although the experiments agree better with the results derived from (1) than those obtained from (2), there seems to me to still be rather significant deviations. Do the experiments give any hint of how one might improve the model beyond (1)? Or do the authors have some intuition concerning this? In general I believe that experimental hints on how we could improve the model could give an added value to this study.

As a very minor remark, in equation (2) I presume that the parenthesis on the left hand side of the equality sign does not play any role and could be removed.

Reviewer #3 (Remarks to the Author):

This paper discusses the experimental realization of a quantum absorption refrigerator with 3 trapped ions. The paper aims to compare the cooling capability when using thermal states or squeezed states, and the performance in the single shot regime with the steady state limit.

The results show that cooling is most effective when there is no squeezing at all. It also shows that when the interaction is switched off at the right moment, so that the system does not reach the steady state, the absorption fridge cools quicker and stronger than in the steady state case.

This is an exciting experiment that will be valuable for the quantum thermodynamics community in testing one of the much discussed heat machines, the quantum absorption refrigerator, for the first time experimentally.

I found the paper clear in the story it is explaining, but had significant problems understanding how the theoretical concepts are realized in the experiment and what the experimental data shown in the figures indicate.

Therefore I recommend that the manuscript is potentially suitable to be published in Nat Comm, but the authors need to revise the manuscript substantially to make it accessible and scientifically sound.

Specific queries are the following:

Eq. (1): the Coulomb interaction is a long-range binary interaction. Further details of how the trip-linear interaction comes about and how bi-linear interactions are avoided in the experiment are required.

Also a clearer exposure in the main text explaining what constitutes the 3 ion modes, that make up the absorption refrigerator, is needed. (it's in the methods, but should be included in the main text)

Fig 2: comparison of data with Eq.(2) shows deviations, as mentioned in the text.

But why should (2) hold in the first place and what is the physical reason it doesn't actually?

Or alternatively, maybe (2) shouldn't hold, why not?, and then why is there a point comparing the experimental data with it?

page 5: Could the authors add a sentence explaining why $E_c=E_w=E_h$?

page 7: it says "we next investigate ... quantum mechanical coherence ... we prepare a squeezed thermal state": what are the coherences of this state? The subsequent discussion mentions that the mean photon number changes as a result of squeezing. So isn't that the main effect of squeezing/in what sense is there a link to coherence?

Fig 4. contains an incomplete sentence referring to (2).

the word "nett" probably should be "net" throughout

Methods:

page 14: Why do the authors chose to first cool (sideband) and then heat again?

(I like the sweat drops, sunglasses and hats of the work, warm and cold modes, respectively, in Fig 1.)

We thank all Reviewers for the careful reading of the manuscript and useful suggestions. In response to the Reviewers' comments we made substantial changes to the manuscript. In particular, we clarified claims and significantly expanded description of the experimental setup and procedures in the main text of the manuscript and in the Method section. We also added several new chapters to the Supplementary information with more technical details related to experimental parameters, data analysis and error propagation.

The reviewers expressed concerns whether the refrigeration can be achieved in our system using only the unitary evolution, and how the experiment fits in the general context of quantum thermodynamics. To address these concerns, we have expanded the discussion in main text and in the Methods section of the manuscript. Since the detailed analysis of underlying theoretical concepts is beyond the scope of the present experimental work, we have also published a complementary manuscript (see Quantum 1, 37 (2017)) where these ideas are discussed more extensively.

The other big issue that was solely expressed by Reviewer 1 was the validity of the data evaluation method. In our reply, we note that our method is merely a generalization of the common thermometry method used in the experiments with trapped ions. As we point out in the reply, this method does not lead to unphysical or biased values as suggested by Reviewer 1. We also show that, while the systematic drifts of the trap frequencies can introduce the systematic error, such errors are insignificant for the trap frequency instabilities that we have measured in the experiment. We are thus confident that our data evaluation method is valid and gives an unbiased estimate of the motional state energy.

We also have re-processed the data to address the other suggestions of Reviewer 1. The new analysis now includes contribution of spectator modes and decay of the coherence in the motional modes. We found that contribution of the spectator modes has a negligible effect on the data. However, including phonon coherence decay in the data analysis caused small changes in the data shown in the figures 2-4 (by a few percents on average). In addition, we followed suggestion of the referee and changed the procedure to extract the equilibrium mean phonon number of the cold mode for the figure 2e, which led to changes in this figure. We also found that the vertical error bars previously shown in the Figure 2e were indeed underestimated, as the referee has pointed out. We are grateful to Reviewer 1 for his comments that led to the improvements in the data analysis.

We believe that we have addressed all the comments and suggestions of the reviewers, and the manuscript can now be reconsidered for publication in the Nature Communications. Below we give detailed response to the referee's comments and suggestions. Our response is typed in blue, and the original referee comments in black.

Reviewers' comments:

Reviewer #1 (Remarks to the Author):

In their manuscript „Quantum absorption refrigerator with trapped ions“, Maslennikov et al. describe an experiment where three collective vibrational modes of an ion crystal are controllably tuned to a three-mode resonance, such that a sum frequency generation process can take place, where two phonons of a “work” and “cold” mode are annihilated and a phonon in a “hot” mode is created. The authors prepare the modes in thermal states with predefined mean phonon numbers, and measure resulting steady-state mean phonon numbers as well as the temporal evolution. In addition, the authors investigate the case where the “work” mode is prepared in thermal squeezed states, and investigate the effect on the system dynamics and steady-state. Furthermore, the authors demonstrate a “single-shot” cooling technique, where the time the system spends on the resonance is chosen to enforce maximum depletion of the cold mode.

Certainly, “quantum thermodynamics” is a timely subject, with a massive imbalance between theoretical and experimental work, and the idea of using a three-ion crystal with a nonlinear resonance as an experimental platform for studying refrigeration in the quantum domain is compelling. However, careful

assessment of this work leads to concerns about the underlying concepts, the evaluation of the data, the question whether the data supports the claims made by the authors, and the interpretation. I list my main concerns:

-Unitary refrigeration? This work deals with closed system with three degrees of freedom, far away from the thermodynamic limit. Switching on the interaction Hamiltonian Eq. 1 causes a unitary evolution, which cannot change the entropy of the entire system. Therefore, can a “steady state” be achieved at all, where all modes are in thermal states? Would the process not be reversed at some point without additional non-unitary processes, and the “cold” mode would assume its initial state? Why does something like a steady-state occur here? This issue could probably be addressed by an analysis showing that the Poincare recurrence time by far exceeds any other timescale of the system.

The authors vaguely address this issue in the paragraph at l74 and in the methods section l140, clarification is however not achieved.

The referee points out correctly that the unitarily evolving system never truly converges to a steady state. However, in stark contrast to qubit-based absorption refrigerator studies, we do observe that an effective equilibration of the single-mode energies takes place; they rapidly approach the values associated to the infinite time average of the unitarily evolving state, which is identical to the fully dephased counterpart of the initial thermal state. We attribute this effective equilibration (which, by the way, would not be observable for temperatures very close to the quantum ground state) to the broad incommensurate spectrum of energies of the trilinear interaction Hamiltonian; many of these energies are occupied for the initial temperatures considered here. Indeed, we could not observe any sort of recurrence of the initial single-mode energies in our simulations, even after a very long interaction time several orders of magnitude larger than the inverse coupling strength. Peak amplitudes of the residual oscillations decrease rapidly with the initial temperatures, i.e. the populated effective dimension of the problem.

This is a remarkable feature, given also that the corresponding classical system of three oscillators has integrals of motion and can even be solved exactly for each set of initial conditions. Nevertheless, effective equilibration around the infinite time average is observed in both the quantum and the classical model. A discussion of the effective equilibration feature and the classical model can be found in a theory manuscript by some of the authors that we now cite and that is meant to complement the present work (see <https://arxiv.org/abs/1709.08353>, Quantum 1, 37 (2017)). The details presented there are beyond the scope of the present work, but we have added clarifying sentences to the second paragraph of the Results section and at the last paragraph of “Equilibrium and steady state populations” in the Methods section (including the citation). Notice that we distinguish the “asymptotic steady state” from the true “equilibrium” that occurs if the initial temperatures fulfill equation 2.

Furthermore, the authors seem to use “steady-state” and “equilibrium” interchangeably, this should also be clarified.

We repeatedly checked the manuscript for consistent use of the terms “steady state” and “equilibrium”. We stress again that we make a clear distinction between both terms, as discussed in the first part of the Methods section (“Equilibrium and steady state populations”). We now consistently refer to the *asymptotic steady state* when we speak of the infinite-time average values around which the coherently evolving single-mode energies *effectively* equilibrate. The term equilibrium, on the other hand, is exclusively used to refer to an initial thermal state whose parameters fulfill Equation (2) and that is stationary under the coherent evolution as it commutes with the Hamiltonian (1).

Concerning the “single-shot cooling” method, it is hard to judge how this can be linked to actual refrigeration, as this is a finite-time unitary operation. If we imagine a persistent heat load on the “cold”

mode, would it be possible to apply such “single shot cooling” repeatedly to maintain the “cold” mode at a low temperature?

I would like to illustrate this with an example: If we prepare a vibrational mode of an ion crystal in a thermal state, drive a rapid adiabatic passage state transfer on the red motional sideband, and subsequently monitor and analyze Rabi oscillation on the blue motional sideband, we would observe a decreased “temperature”. However, it is obvious that the rsb state transfer is a unitary operation rather than cooling.

It is a good example, since the red sideband (RSB) operation the referee describes forms part of the well known sideband cooling method (in most of the implementations rapid adiabatic passage is usually replaced by the red sideband evolution). While the entropy of the whole system did not change, the entropy (and temperature) of the motional mode alone has DECREASED as the result of such RSB operation.

To fully implement the sideband cooling in the referee’s example, one has to optically pump the internal state of the ion back to the original state after the RSB transition or, using the thermodynamic language, couple the spin degree of freedom to the zero temperature bath. This is the same operation that one has to do in the referee’s example before driving the blue sideband. Otherwise the spin state is entangled with the motional state of the mode (after the rapid adiabatic passage pure initial state $\sum_{n=0}^{\infty} \alpha_n | \downarrow \rangle | n \rangle$ will evolve into $\alpha_0 | \downarrow \rangle | 0 \rangle + \sum_{n=1}^{\infty} \alpha_n | \uparrow \rangle | n-1 \rangle$, similarly for the mixed state), and traditional analysis of the Rabi oscillations can not be applied for such initial state to measure the mode energy. In the referee’s example, the implicitly assumed optical pumping step is, obviously, a non-unitary operation and decreases the entropy of the whole system -- contrary to the referee’s claim that his example describes only unitary evolution.

Similarly, in our experiment we have implemented the unitary part of the cooling process. Coupling to a bath is simulated in the beginning of the cycle by preparing different initial states of the ions. In “Equilibrium and steady state populations” of the methods section, we mention that this yields a valid approximation of the refrigerator dynamics in the limit of weak coupling, i.e. slow thermalization. Full implementation of simultaneous interaction and coupling to the bath is the subject of the future experimental work.

To link the single shot cooling to the refrigeration process, we note that the aim of the single shot cooling is to prepare a set of oscillators in a state with the lowest possible temperature in a short time. [see Ref 24]. For example, imagine the cold, hot, and work bodies each consist of N harmonic oscillators with temperatures T_h , T_w , T_c . From the N harmonic oscillators, we take one oscillator from each body (such that we have 3 oscillators to work with), put them in our “single shot cooler”, let them interact as described in the manuscript until the the lowest energy of the cold oscillator is reached, and then put these oscillators aside. We do it N times for all the oscillators in the heat bodies. After that we recombine the corresponding oscillators back to the cold, work, and hot bodies and let them thermalise. The achieved temperature of the cold body in this case would be lower than temperatures achieved by the classical absorption refrigerator for the same initial conditions.

It would be possible to use single shot cooling to remove the heat from the cold body continuously, by iteratively returning oscillators back to the heat bodies and letting them thermalise after each single shot. However, in this case, the minimum temperature that one can reach is the same as in the classical case, because the conditions for the single shot cooling are the same as for steady state cooling (equation 4).

The following sentence was added to the main manuscript:

“However, to demonstrate the advantages of this method in practice, one would have to implement the trilinear interaction between the modes and couple each of the modes to its corresponding bath.”

-Data evaluation: I am wondering about the small vertical error bars in Fig 2 a-d. E.g., for panel d) at $\bar{n}_c^{(in)}=0.5$, the horizontal error is about ± 0.036 , while it is only 0.013 for the hot mode mean phonon number *difference*, which results from *two* measurements.

We thank the referee for the opportunity to clarify the error propagation procedure.

In his comment, the referee compares the error bar of the *absolute* value of the *cold* mode with the measurement of the *change* of the the \bar{n} for *another, hot* mode. It is not a fair comparison since: **a)** different methods are employed to determine these values, **b)** different number of experimental shots was used in these measurements, and **c)** some systematic errors that affect the *absolute* values are cancelled out when we measure the changes in mean phonon numbers.

In particular, *absolute* values of the mean phonon number in the main text of the manuscript are always determined from calibration of the preparation procedure. This calibration procedure relies on the measurement of the Rabi oscillations and extracting the mean phonon number (and squeezing parameter in the case where the squeezing is used) from the evolution of the blue motional sideband. [See Supplementary information, “Calibration of experimental initial conditions”]

The *change* of the mean phonon number before and after interaction is always determined in the main text of the manuscript from the change of the ion brightness after a red sideband is driven with a fixed pulse duration. [See Method section]. Such measurement procedure allows us to eliminate or greatly reduce influence of several systematic effects that contributes to the absolute value of the mean phonon number. (For example influence of the spectator modes, imperfections of the state detection, etc.).

This led me to look at the paragraph “Motional state detection” in the Methods section. Here, the authors state that rather than recording Rabi oscillations on one or more motional sidebands and extracting the required mean phonon number, they one probe the red sideband of the respective mode at a single fixed time t_{rsb} at a net Rabi frequency Ω . Both parameters are not indicated. It is also not explained if t_{rsb} is chosen differently for different modes or particular measurements.

We thank referee for this remark, Indeed the values for t_{rsb} and Ω were accidentally omitted in the initial version of the manuscript. The t_{rsb} parameter is chosen such that the probability of finding the ion in the bright state is most sensitive to the variations of the mean phonon number around the value of interest. For the “hot” mode (Figure 2), where typical number of phonons in the mode is low, we chose $\Omega t_{rsb} = 2\pi/3$, for the “cold” mode, where the mean phonon number is higher, we use $\Omega t_{rsb} = \pi/3$.

We have added the used values of t_{rsb} in the Methods section of the manuscript as well as the (vacuum) Rabi frequencies of the sidebands in the Supplementary Material Table S2.

The authors link some observed probability to measure the ion in “spin up” after driving the sideband for t_{rsb} via Eq. 6. As the phonon distribution $p(n)$ is not given, the authors precompute some expected distribution, leading to an expected mean phonon number \bar{n}_{th} , and use the measured probability $p_{\uparrow exp}$ to obtain a corrected “experimental result”.

I have concerns about the validity of this approach. If the measurement would contain no information about the desired quantity, we would have $\frac{\partial \bar{n}_{th}}{\partial p_{\uparrow th}}=0$, and the Eq. 7 would merely return the precomputed value. In other words, the results are biased towards an expected outcome. It is also unclear how the authors obtain the shown confidence intervals for measured mean phonon numbers.

We thank the referee for the opportunity to explain this method in more details. It indeed deserves more detailed justification.

The method that we use is a generalisation of the well known procedure to determine the mean phonon number in the ion's mode of motion from the asymmetry of red and blue sidebands [Phys. Rev. Lett. 75, 4011 (1995)]. In that case ratio of probabilities to drive the spin flip transition on the red and blue sidebands can be related to the mode mean phonon number, **assuming** the phonon number distribution is thermal. If the state of motion is indeed thermal, this method gives an unbiased estimate of the phonon number.

In our experiment the state of motion can deviate from thermal, and information regarding the phonon number distribution of the corresponding state is required for converting the probability of the sideband transition to the mean phonon number. The phonon number distribution can be measured; we did it to calibrate the preparation procedure for the initial states. However, this process is time consuming and susceptible to systematic errors in the state detection, to the presence of spectator modes, to fluctuations of the power in the raman beams, etc. And it gives a relatively large uncertainty for the final result.

We note however that, in this experiment, we always want to know the *difference* between the mean phonon number in the initial and final states with high precision, not the absolute mean phonon number. As we already mentioned above, several sources of systematic errors (state detection error, contribution of spectator modes, etc.) that contribute to the uncertainty of the absolute value of the mean phonon number cancel out when we measure the change of the mean phonon number.

For the measurements shown in Figs. 2-4, the expected phonon number distribution for the mode of interest was instead calculated by simulating the unitary evolution of the system. Knowing this distribution we determine the expected mean phonon number as the function of the spin flip probability $\bar{n}(p_{\uparrow})$ and the gradient $\left(\frac{\partial \bar{n}}{\partial p_{\uparrow}}\right)$ and use the difference between the observed and expected ion brightness to estimate the mean phonon number in the mode [See equation 8 in updated manuscript].

This method gives the correct result if we know the phonon number distribution exactly. The most likely reason for the phonon number distribution evolution to deviate from the expected one is that the resonance condition $\omega_h = \omega_w + \omega_c$ is not satisfied precisely, and the evolution of the system brings us to a different phonon number distribution. We have verified that our method is robust to this error within the uncertainty of the trap frequency drifts. As shown in the plot below the gradient $\left(\frac{\partial \bar{n}}{\partial p_{\uparrow}}\right)$ changes by less than 3% if the detuning between the modes does not exceed 500 Hz, and is much smaller for the typical detunings in our experiment (~ 200 Hz). This gives negligible contribution to the difference of the mean phonon numbers, if compared to the other sources of the experimental uncertainties (e.g statistical uncertainty of p_{\uparrow})

The uncertainty of the mean phonon number is calculated as $\Delta p_{\uparrow} \left(\frac{\partial \bar{n}_h}{\partial p_{\uparrow}} \right)$, where Δp_{\uparrow} is the uncertainty of the probability of the spin-flip transition.

The extreme case where $\left(\frac{\partial \bar{n}_h}{\partial p_{\uparrow}} \right) = 0$, mentioned by the referee, does not occur in our system. ($p_{\uparrow}(\bar{n})$ is a continuous function and its derivative cannot go to infinity. Indeed, the height of the red sideband can not change suddenly with infinitely small changes of the mode energy for any physically relevant distribution of the phonon numbers.) As an example, we plot $p_{\uparrow}(\bar{n})$ for the thermal states for a rsb pulse duration of $t_{rsb} = \pi/3$ and $t_{rsb} = 2\pi/3$ below.

In the opposite extreme case, $\left(\frac{\partial \bar{n}_h}{\partial p_{\uparrow}} \right) \rightarrow \infty$, which is indeed possible, uncertainty of the measured phonon number would be infinite, as one would expect from the measurement that does not provide information regarding the desired quantity.

We took specific care to avoid such extreme cases. As we already explained above, duration of the red sideband pulse t_{rsb} was chosen to minimise $\left(\frac{\partial \bar{n}_h}{\partial p_{\uparrow}} \right)$ and, therefore, minimise uncertainty of the mean phonon number \bar{n} . The $2\pi/3$ pulse was chosen for the measurement of hot mode \bar{n} where the expected mean phonon number is typically less than 1, and $\pi/3$ pulse for measurements of the cold mode, where expected \bar{n}_c was larger (between 2.1 and 3.1). In any case the true value of the mean phonon number lays within the confidence interval with high probability. We cannot find a realistic scenario where this is not the case and, therefore, confident that our results are within the uncertainties that we have specified.

We thus conclude that the referee's concerns regarding validity of our approach are not justified.

In response to referee's comments and to further clarify the validity of the approach, we added the following sentence to the section "Motional State Detection":

“We have tested that the value of the partial derivative in equation (8) is robust to the choice of δ in equation (8), and to the changes of the phonon number distribution due to variations of the detuning Δ from resonance ($\partial \bar{n}_{th}(\tau)/\partial p_{1th}(\tau)$ changes less than 1% for $\Delta = 200$ Hz).”

Furthermore, the coupling to spectator modes while driving the red sideband is not taken into account in Eq. 6. While the beams R1, R2, R3 are shown in Fig. 1, the authors do not explain which beams are used for coupling to which mode. Using R2 and R3 for driving the rsb of the “cold” mode on one of the outer ions, one would expect parasitic coupling to the “work mode”, .i.e. the resulting mean phonon number on the “cold” mode would depend on the state of the “work” mode. These issues raise concerns about the validity of all measurement data presented in this work.

The coupling between the work and cold modes is small since these modes are separated from each other by about 300 kHz and the maximum Rabi frequencies for all the sidebands does not exceed ~10 kHz. [See Supplementary Information for more details and the figure below that shows typical sideband arrangement]. Moreover the data presented in the Fig 2-4 is the difference of two measurements, in which one measurement is done after the trilinear interaction was turned on, and another one after preparation of the initial state (without turning on interaction). Since the number of phonons contribution of the spectator modes is almost the same for both cases, contribution of the spectator mode to the *change* of the signal largely cancels out.

For the Figure 2 this cancellation is exact, since the “hot” mode can couple in this way only to other axial modes that are always cooled to the ground state of motion throughout the experiment. For Figs 3 and 4 this cancellation is not perfect, since the number of phonons in the work mode and phonon statistics can change after the refrigerator evolution.

Following suggestions of the referee we did a more careful analysis of the spectator modes contribution. The contributions of the spectator modes to the main results displayed at Figures 3 (g, l) and Figure 4 are presented in Table S3 of the Supplementary Materials section. The largest systematic error contributes approximately $2.5 \cdot 10^{-4}$ phonons to the measured mean phonon number in the cold mode. We also calculate and propagate this error into every point of the “evolution curves” (Fig 3(a-f and h-k)).

Changes:

1. Added Supplementary Table S3 displaying the estimates of the spectator (work) mode contributions to the measured mean phonon numbers displayed in Figures 3 g,l and Figure 4.
2. Added Supplementary Figure S5 displaying the spectrum of the “cold” and “work” mode for illustration of the spectator mode contributions.
3. Added outline of the estimation procedure into “Error Analysis” section in the Supplementary Material.

We believe that all effects related to spectator modes on the state detection are now properly analyzed, and concerns of the referee regarding validity of the data are not justified.

-Quantum coherence: The authors claim to study the “influence of quantum mechanical coherence” by preparing the “work” mode in thermal squeezed states. The motivation, results and interpretation are rather unclear to me. It seems that the squeezing operation merely increases the mean phonon number of the “work” mode, which leads to increased depletion of the “cold” mode. The question whether the cooling efficiency is increased or not as compared to a thermal state of the same effective mean phonon number cannot be answered on the basis of the data presented in Fig. 3, in contrast to the claim in I95-I97.

It also remains unclear if the discrepancy in the simulations (green areas in Fig 3. G & I) is actually due to quantum coherence in the work mode, or merely due to different higher momenta in the phonon statistics.

The referee is correct that the squeezing of the work mode increases the mean phonon number of the work mode. It is interesting to note, however, that the entropy of the work mode remains the same as before, since the squeezing is the unitary operation. Classical thermodynamics would allow no cooling in such case.

There are two effects that we wanted to show by experimenting with squeezed states. The first one is that just by squeezing the initial thermal state of the “work” mode, one can cool the “cold mode”. The data shown in Figure 3 I obviously shows that it is true.

How effective is this cooling is the other question that we have tried to address by comparing the results to those obtained with thermal states in Figure 3 g. Our experimental data shows qualitative agreement with numerical simulations. The numerical simulation, in turn, clearly indicate that the cooling efficiency is lower when squeezed states are used. We have revised the manuscript to make these claims and logic more transparent.

The authors do not explain why they use squeezing rather than displacement, which is easier to achieve and also serves to increase the work mode mean phonon number. I suspect that squeezing was chosen because it sounds more “quantum”.

Since we never had the intention to use displacement nor did we at all considered it, we did not explain why we chose squeezing instead of displacement in the original version of the manuscript. We are grateful to the referee for raising this questions and for the opportunity to clarify this issue.

First of all, our work addresses a thermodynamic scenario of energy flows between (stationary) thermal reservoirs, not between time-dependent external driving fields that would cause displacement by means of a classical source of work. Second, we then also considered squeezed thermal states to measure the

influence of external quantum resources. This explicitly addresses the well noticed claims made in numerous theoretical works, e.g. J. Rosznagel, et al, *Nanoscale Heat Engine beyond the Carnot Limit*, Phys. Rev. Lett. 112, 030602 (2014), L. A. Correa, et. al, *Quantum-Enhanced Absorption Refrigerators*, Sci. Rep. 4, 3949 (2014), G. Manzano, et. al, *Entropy Production and Thermodynamic Power of the Squeezed Thermal Reservoir*, Phys. Rev. E 93, 052120 (2016), W. Niedenzu, et.al, “*On the operation of machines powered by quantum non-thermal baths*”, New J. Phys. 18, 083012 (2016). These works explore squeezed thermal reservoirs as a quantum resource that can increase thermodynamic efficiencies to even beyond the Carnot limit.-We note however, that such statements are a subject of intense debates [see for example Phys. Rev. E 92, 042126 (2015)], and a final interpretation is far from being established. Nevertheless such claims have already inspired some experimental work in this direction [for example, Phys. Rev. X 7, 031044 (2017)]

We do not intend to take sides in these debates and insist on the particular interpretation of our data. Instead, our goal here is to put the debate on solid ground by providing sound experimental data.

-Data quality & claims: The quality of the data seems to be insufficient to support some claims made by the authors. For Fig. 2 a-d, the authors state “The numerical simulation agree well with the experiment.” This is certainly not the case for panel b, which is not discussed.

Fig. 2b discrepancy: We agree to the referee that 33% of the points on the Fig 2b are outside of theoretical prediction range. The systematic errors in our experiment are mostly coming from the slow frequency drifts of the radial modes. They arise from the common situations such as air-conditioning flow intermittency and slow temperature fluctuations that add unwanted offsets to our trap frequency stabilization control loop. Typically, we check the radial mode frequencies in between each point presented on the curve (1-2 hours time interval). If the mode is found to have drifted away from the set value, we adjust the DC rod voltages and check whether resonance condition is fulfilled by measuring avoided crossing between motional modes. We note, that we mostly able to notice such drifts before they affect our experimental data, and conjecture that the deviation seen in panel b is the result of such sudden drift. We further note that only 5 points out of 23 on Fig. 2 (a-d) have significant deviation from the theoretical value, which is well within the expected range assuming normal distribution of the measurement errors.

We are thus confident that the main result of this measurement (Fig 2e) is unaffected by this displacement. In addition, we follow standard experimental practice and present all the data that were collected during particular experimental run. We see no reason to exclude any points that were measured in experiment.

The data in Fig. 2e results from a very gross evaluation method of connecting two points around the vertical “equilibrium” lines in panels a-d. From visual comparison with panels a-d, the vertical error bars in e seem to be substantially underestimated.

Fig. 2e evaluation:

Following the referee’s remark, we have employed a different procedure to determine $n_c^{(eq)}$ shown in Figure 2e. The new procedure takes into account each point in the dataset presented at Fig 2 a-d. We have numerically computed a mean phonon number difference ε_h in the “hot mode” for 61000 different initial conditions using Hamiltonian Eq. (1) and constructed the fitting function by interpolating the results of these numerical simulations.

From the fit, we obtain the values of initial “hot” and “work” mode numbers and use them to find the n_c that corresponds to the intercept point where $\varepsilon_h = 0$ and its uncertainty. We believe that the data is properly analyzed now. We updated Figure 2e, its caption and added a paragraph in the Supplementary Materials that discusses the procedure and details of the error propagation analysis.

Indeed, the previous data analysis procedure underestimated the vertical error bars of Fig 2e. We thank the referee for this comment. We note however that the data in the new Figure 2e still supports the main claim (observation of the absorption refrigeration), and this claim is not altered by the change of the error bar size or shift of the data points with respect to the old version of the Figure 2e.

In Fig. 3 panels a-f,h-k, the data exhibits only a very vague tendency to have something to do with the simulations, which supports doubts about the data evaluation method.

Fig 3. a-f, h-k: Assuming normal distribution of the experimental errors, 68% of the experimental points should be within 1 standard deviation from the theoretical simulations. In the data presented in the panels a-f, h-k 62% of points satisfy this criteria, which supports validity of our evaluation method and error propagation procedure.

Few points in the panels h and I indeed look off, and we attribute it to the instability of the trap frequencies as discussed above, which have particularly large effects on preparation of the squeezed thermal states. However, as before, we saw no reason to discard any data and have included all of the experimental points in the figure.

Note that the theoretical simulations shown in these panels do not include the uncertainty of the initial mean phonon numbers, like we do for the bottom panels (g and I). Including these uncertainties would make the overlap between the theoretical curves and experimental points better.

The data presented in panels g & I does not allow to draw any conclusion on different performances of thermal and squeezed thermal states.

For comparison of data between panels 3g and 3I, we believe that one conclusion can be made: within our measurement accuracy, there is no visible improvement in cooling performance when the squeezed number thermal states are used in the work mode. Moreover, we note that our experimental points qualitatively agree with numerical predictions. The numerical simulations presented there support the statement that the cooling efficiency is lower when squeezed states are used. As we mention in the reply above, we have changed the concluding sentence of this paragraph to clarify our claims.

Why is there no data point in panel g for $\bar{n}_w > 5$, although there is one in Fig. 4?

The data presented on the Figs 3 and 4 are separate experimental runs that employ different experimental sequences and acquisition methods. All the available experimental data are presented on the Figures. There is no data point with $n > 4.4$ on Figure 3, because no experimental data are available. In fact, we intentionally avoided using high mean phonon numbers in Fig 3, since reproducible preparation of motional states with high mean phonon number over long data runs (days) that are required for Fig 3 was challenging, due to trap frequency drifts described above. In contrast the experiment in the Figure 4 requires the measurements of only 2 points and was done much faster (hours). Stability issue did not arise here.

For thermal states of the “work” mode, only one data points supports the general observation of cooling with significance. This cooling amounts only to less than 10% of the initial thermal energy of the “cold” mode.

The motivation of experiments presented on Fig.3 was to study the absorption refrigerator in different regimes, and to observe transition from heating to cooling regimes as the temperature of

the work mode increases. It is indeed true that one point in this experiment shows significant cooling. However, as referee has pointed out, experimental data support the observation of cooling with significance and agree with the theoretical predictions.

We also would like to point out that the absolute value of the cooling effect is irrelevant here, as long as it has been reliably demonstrated. For example, commercial refrigerators that authors have in their kitchens reduce the thermal energy of stored beer by less than 10% (from 300K to about 277 K), which makes the drink much more enjoyable and the refrigerator a very useful device. The goal was never to set the world record on the lowest temperature. Instead the goal of this study was to demonstrate the absorption refrigerator in the trapped ion system. We believe that this goal has been achieved.

We would like to note also that the general observation of cooling by the quantum absorption refrigerator (in different regimes) is also supported by the data presented in Fig. 2 and Fig 4 and is well justified.

In conclusion, this could be an interesting work if it would be more thorough on analyzing and discussing the validity and applicability of the underlying concepts, and if the data acquisition and evaluation would be more trustworthy. In the present state, this manuscript is not suitable for publication in Nature Communications.

I list a number of further issues below:

Fig. 2e: The authors indicate a “no cooling” region. Would the region above $T_c = T_h$ not also be a “no cooling” region?

The referee is correct. Yes, the region above $T_c = T_h$ is also a “no cooling” region. In fact, it is rather a “heating” region, which can be reached by decreasing the “work” mode temperature. To avoid further confusion, we have removed the shaded region from the plot.

We updated Figure 2 and removed green shades area.

p7, l78: The statement leading to Eq. (3) seems to be logically disconnected from the text before and after it.

We have added a sentence that we think provides the logical transition: “If the energy of a mode in the asymptotic steady state is lower than its initial energy, the mode is cooled down. For observation of cooling in the “cold mode”, the initial energy of the work mode must satisfy”

We also removed the sentence leading to Eq. 3: “For cooling the cold mode ($\epsilon_c < 0$), the following inequality must be satisfied”

Fig 3. l): The blue shaded area is supposed to pertain to Eq. 2, which however holds only for thermal states. The discrepancy with respect to panel g is not explained.

Fig 3, caption: The statement “A slight difference...” should be put into some context.

The blue shaded area is a benchmark predicted by simple arguments from classical thermodynamics. Given an initial state with temperatures out of equilibrium, the formula gives the relevant thermal equilibrium that the mode energies could in principle reach at fixed values of the conserved quantities under the Hamiltonian (1). While the formula was derived for the thermal equilibrium states, one might also assume it to be valid for the average mode energies of a non-thermal (steady) state. (In fact, several authors obtain consistent description of the squeezed thermal bath by introducing an effective temperature, see for example Ref. 18). We prefer to leave

the blue line in the Fig. 3 intact to show explicitly that the long-time average state that the coupled system evolves into is not thermal.

Discrepancy of panel I with respect to panel g: The blue shade in both Figures are calculated for different initial mean phonon numbers in both the hot and cold mode. For the case of Figure 3g it is given by the calibration value that we assume is persistent throughout the experiment. For Figure 3h-k, the initial mean phonon number is measured independently for each branch and the final value used in Figure 3l is given by the average of the numbers presented for each branch. This value of hot and cold of 0.49(3) and 2.78(17) is slightly different from 0.66(4) and 2.63(17) used for the thermal states plots, which results in the observed discrepancy of the blue shaded areas. However, this does not affect the main conclusion drawn from the green shade behavior together with experimental points.

In response to the referee's remarks, we have added the following line to the caption of Figure 3 for clarification:

"For **h-k**, initial mean phonon numbers of hot and cold modes were measured before each experimental run. Taking the average value of $\bar{n}_h^{(in)} = 0.49(3)$ to calculate the theoretical predictions of **I** mainly shifts the green shaded line vertically compared to $\bar{n}_h^{(in)}$ of **g**."

p11, I118: the comparison with an actual refrigerator would be suitable for an entertaining press release or feature article, but not for a scientific publication. The authors compare a *part* of a classical refrigerator to their system, where the three ions comprise the coolant, the material to be cooled down, and part of the machinery. Here, one could debate if the trap could be seen as the 'compressor', as it is required to tune the crystal to the parametric resonance. Then, the quantitative comparison in terms of mass would lead to a very different result.

We would like to point out that similar comparisons were included in the scientific papers in highly respected journals by other authors [see, for example, Science 352, 325 (2016)]. However, following the suggestion of the referee, we have decided to remove this comparison from our manuscript.

p11, I125: "...hence demonstrating that squeezing could be used as a quantum fuel" contradicts somewhat with the statement on p9, I96 "...which implies that squeezing of the work mode decreases the cooling performance".

We see no contradiction here. We can use the squeezing as a resource (or quantum fuel), but it would be a little bit more efficient to prepare the thermal state instead of spending the same energy on preparation of the squeezed thermal state.

Eq. 4: It is not clear to me if this equation (along with the second law below) holds for the system, which is far away from the thermodynamic limit. Extracting/adding heat (\dot{Q}) from/to a finite-sized system will substantially change its temperature. A constant temperature at finite heat extraction/addition is only maintained for an infinitely large system (a so-called reservoir).

The second law of thermodynamics is a universal law of nature that holds for finite systems as well. A dynamical version of the second law for open quantum systems can, for instance, be found in R. Alicki, J. Phys. A 12, L103 (1979).

Equation 5 (previously 4) holds in the steady-state scenario of an ideal adiabatic absorption fridge where each mode is in contact with its own thermal reservoir at temperature T_i . In this open-system model, the second law holds with the heating rates \dot{Q}_i given by the rates of change of the mode energies due to the dissipative bath coupling, and with T_i the temperatures of the baths. Given the trilinear exchange interaction, Eq. (5) yields the condition (2) for the bath temperatures (i.e. in our case, initial temperatures before interaction) at which the heat flows balance in such a way that the system entropy does not increase, see e.g. Ref. 8.

l136: here, Boltzmann statistics rather than Bose Einstein statistics apply, the formula however is correct.

This is indeed a matter of terminology. The Boltzmann statistics of energies in a quantum harmonic oscillator means the same as the Bose-Einstein statistics of phonon numbers (quasi-particles without chemical potential) occupying that mode. To avoid unnecessary confusion, we now refer to the “canonical expression for the mean phonon number”.

l140: It is unclear what “The corresponding quantum state” refers to.

We have edited this sentence for clarity:

“The corresponding quantum state $\rho = \rho_h \otimes \rho_w \otimes \rho_c$ is stationary as it commutes with the interaction Hamiltonian (1)”

is replaced by:

“The quantum state $\rho = \rho_h \otimes \rho_w \otimes \rho_c$ corresponding to the equilibrium condition (2) is stationary as it commutes with the interaction Hamiltonian (1).”

l145: “...obtained by dephasing the initial ensemble..” It is unclear if “dephasing” refers only to a computation, and if yes, if some actual dephasing takes place on relevant timescales, how it is caused, and what its implications would be. Furthermore, the implication of the correlations is unclear. The entire paragraph is rather confusing.

We have revamped this paragraph of the Methods section to clarify the distinction between a stationary equilibrium for specific initial conditions and the effective equilibration of non-stationary initial energies around an asymptotic steady state given by the infinite time average over the unitary evolution. To comply with the referee’s suggestion, we now make clear that this infinite time average “can be computed” by dephasing the initial state. No actual dephasing takes place on the relevant time scales in the experiment.

l148: also unclear: the timescale for “the fast internal refrigerator dynamics” should be ξ . The solid lines in Fig. 3 indicate that what the authors seem to call “thermalization” happens on the same timescale, with a factor ~ 2 difference. The statement implies a timescale separation, and it implies some sort of control parameter.

We took the referee’s suggestion into account and extended the paragraph mentioning thermalization with external reservoirs as opposed to the fast internal dynamics. No thermalization occurs in the present experiment, and the effective equilibration we observe (e.g. in the solid lines in Fig.3) is *not* thermalization, but an intrinsic feature that occurs on the fast internal time scale $1/\xi$ of the unitary dynamics. In the paragraph we now make the point that the studied scenario is a good approximation for what happens when the refrigerator modes are in contact with thermal reservoirs, but the thermalization rates are much smaller than ξ . In that case, the unitary evolution captures the relevant intrinsic phenomena of single-shot cooling enhancement and fast effective equilibration that would also be present in the case of slow thermalization. In the opposing (and less relevant) case of fast thermalization, the internal dynamics would be thwarted and the three modes would hardly evolve away from their initial energies.

For the simulated time evolution plotted in the manuscript, we also assumed that no dephasing takes place on the relevant time scales. This is indeed the case as we measured a coherence time of the relevant phononic excitations > 8 ms (see explanation below, Methods section “Experimental setup” and Supplementary Table S2).

l160: what is the “coherence time of a single phonon”?

In analogy to the two-level spin system this time describes the dephasing of the superposition states of motion. This characteristic dephasing time sets the upper limit to the unitary evolution time during which one can manage coherent transfer of the motional states between the modes that is employed in our refrigerator. We measure it for each mode with a Ramsey type experiment, using the superposition state between 0 and 1 phonon, thus setting the syntax used in the paper.

To avoid confusion we have changed the sentence to: “Fast fluctuations of the trap frequencies lead to additional decoherence between phonon Fock states; the smallest coherence times for superposition states of the form $(|0\rangle + |1\rangle)/\sqrt{2}$ exceed 8 ms.”

We also added discussion of other sources of decoherence in our system (e.g. heating of the motional modes)

l167: “is always positioned at the edge of the ion chain” how is this achieved?

We now explain it in the manuscript as follows:

“The positioning is accomplished by monitoring the fluorescence of the ions on an EM-CCD camera for ≈ 200 ms between each 100 experimental shots. If the ion jumps to the center, for example due to collisions with background gas, the RF signal sent to the trap is briefly interrupted for a few μ s. This melts and recrystallizes the ion crystal, and is repeated until the bright ion is found at the edge of the chain.”

p14 “state preparation”: it should be note which beam pair is controlling which mode, and the authors should estimate the impact of off-resonant excitation of spectator modes during the preparation of one particular mode. The Lamb-Dicke parameters and the resulting \bar{m} values for each mode should be indicated.

We agree with the comment and have updated the Methods section with a sentence that describes the Raman beam addressing.

To characterize the spectator mode contribution during state preparation process, we performed a series of calibration measurements as described in “Calibration of experimental initial conditions” of Supplementary Materials. We cool all the modes to the ground state of motion, prepare the work mode in a thermal state of different mean phonon numbers and measure the final population of the cold mode. The impact of the work mode preparation on the cold mode is found to be small, as shown in the figure below:

We see that the increase of the population in the work mode does not alter the phonon number in the cold mode. We are thus confident that our calibration procedure of the target mode performed for the vacuum state of the spectator mode remains valid throughout data. However, to minimize the cross-talk between the modes, the mode with largest phonon number was always prepared last during experimental sequence.

We made the following changes:

1. The sentence "The beam is split into three paths R1, R2, and R3, as shown in Fig 1. The R1-R2 pair addresses axial motion, whereas R2-R3 addresses radial motion." is added to "Experimental Setup" in Methods.
2. We have added the above figure as Supplementary Figure S5 together with the description of the experiment in "Error analysis" of Supplementary Materials.
3. Lamb-Dicke parameter estimates for axial and off-resonant radial mode frequencies are added to Supplementary Table S2.

1197: If the low-pass filters are mentioned, also the order and the cutoff should be specified.

The low pass filters used are third order RC filters, with the 3 dB point at 11 kHz. The circuit and the response time measurement using high impedance (1 MOhm oscilloscope input) is shown below.

We have added the description of the filters as: “After preparing the motional modes we adjust the offset voltages applied to trap electrodes via third order RC low pass filters (LPF) with a 3 dB point at 11 kHz.” as the first sentence of the “Energy exchange section in Methods”.

Eq. 6: The choice of η as a contrast is unfortunate, as this commonly denotes a Lamb-Dicke parameter, which is however absorbed into Ω . It should be discussed why spectator modes are not included here.

We agree with the referee and have replaced η with ν in equation (7) of the updated manuscript and the text of the manuscript.

Since the estimated contribution from the spectator modes is almost negligible, the broad discussion together with relevant numbers is added to the Supplementary Materials text (see Supplementary Figure 1 and Supplementary Table S2 and S3).

We also add a brief statement to the end of this paragraph in the Methods section:

“For this choice of detection pulse time, the contribution of adjacent ‘spectator’ modes to the detected signal $p_1(\tau)$ is almost negligible.”

In Eq. S1, the authors include an additional contrast decay, why is this not included in Eq. 6.

Following referee’s remark, we now include the values of the measured contrast decay in Supplementary Table S2 for the modes of interest. We have also included this contrast decay in the data analysis. We have re-processed all the data points and found out that no claims are shattered by these changes. We have updated Equation 7 in the Methods section and all the figures in the manuscript.

The absolute changes in the measured phonon number differences are about 0.0023 on average and are limited to about 0.013 for the extreme cases (point with largest work mode mean phonon number at Fig 4). As an example, we plot the change in the phonon number difference for the case of Fig. 3f due to the addition of the contrast decay as an example of the observed deviations.

The new values for the measured steady states are given in Supplementary Material Tables S4 and S5.

The number of experimental shots per data point is nowhere mentioned.

We have added a new subsection in the Methods section describing experimental sequence in more detail, and included the number of experimental shots per point as well.

Reviewer #2 (Remarks to the Author):

Small thermal machines consisting only of a few degrees of freedom have recently attracted considerable attention, and this manuscript presents an experimental study in this context. The evolution of three resonantly coupled vibrational modes is such that one of these modes are cooled due to the flow of energy between the others, analogous to an absorption refrigerator.

The authors test whether the performance of this device is better when one of the modes is initially in a squeezed thermal state rather than being thermal. They also compare the performance for the long-term evolution of these modes, with a specifically selected short-time evolution. They find that in the long-time regime the coherence in the squeezed state can act as a fuel and induce cooling, but that the performance of the device can be hampered rather than helped by the squeezing. They also find that for the specifically selected short-time evolution, the performance of the device can exceed that of the long-term operation, and be enhanced by the squeezing.

As far as I am aware there is no previous experimental studies of this particular setup in the context of quantum thermal machines, and there are generally still relatively few experiments that probe thermal machines at the quantum level. I would unfortunately not expect that the detailed findings as such would have a significant impact on the field, but the general approach could potentially inspire future experiments and theoretical studies.

Unfortunately, I am somewhat concerned about how the results are presented in relation to the existing notions of thermal machines, but I believe that the authors potentially could improve the clarity of the presentation, as well as the potential for impact, by suitable modifications and additions. Below I try to convey the nature of these concerns, and provide some suggestions.

The absorption refrigerator and many other thermal devices are, both in the standard classical case and in several quantum generalizations, often analyzed in setups where the device is coupled to infinite ideal heat reservoirs that can maintain constant temperatures in spite of indefinite flows of energy through the device. This enables an approach where one analyzes the performance in terms of steady state flows that acts like attractors to the dynamics of the system. In the recent literature on quantum thermal machines, this steady state performance is often compared with the "single-shot" performance, where we select a specific optimal time where the operation of the device is terminated. It appears that the authors aim to study an analogous scenario in their experimental setup.

As far as I understand, the authors do in their main analysis regard the three modes as a closed system that evolves according to the Shrodinger evolution governed by the Hamiltonian (1). In particular I get the impression that they do not add any effects of couplings to an environment (for example via master equations). I would say that such a setting of isolated systems is quite different from a scenario where the systems are attached to ideal reservoirs. In particular, one cannot for isolated systems have steady states in the sense of attractors of the dynamics, and it is in this case also problematic to discuss the evolution of observables in terms of "long time limits". (If the authors indeed wish to claim that some observables display a non-trivial limit in this system, I would urge them to prove this claim, or to cite some source where this is proved.) Unfortunately, the manuscript does repeatedly describe the closed system evolution in terms of "steady states" and "long time limits". Moreover, the whole presentation may give the unfortunate

impression that it almost equates the reservoir scenario with the closed system scenario, and I am afraid that this may cause considerable confusion.

In view of this and the other referees' comments, we have reworded the manuscript, expanded the first part of the Methods section, and included a reference to a theory article by some of the authors that complements the present work (Quantum 1, 37 (2017), see <https://arxiv.org/abs/1709.08353>). We hope that this clarifies the matter. In the present work, we focus on the unitary closed-system dynamics, which is a good approximation for the open-system situation in the limit where the thermalization rates are much lower than the internal dynamics governed by the coupling frequency ξ . In particular, the single-shot cooling enhancement is captured in the unitary model.

Moreover, and in contrast to qubit-based studies of the refrigerator model, the unitary model already exhibits an important feature commonly referred to as *effective equilibration*: On the fast time scale of the internal dynamics, the system energies diverge from their initial values, and they approach and stay close to a value associated to the infinite time average (or completely dephased counterpart) of the coherently evolving state. Note that this effective equilibration around the infinite time average is observed in both the quantum and the classical framework, even though the classical system of three oscillators has integrals of motion and can be solved exactly for each set of initial conditions. A rephasing or recurrence of the initial values is not observed in the simulation, even after very long stretches of interaction time, because of the broad incommensurate energy spectrum of the trilinear interaction Hamiltonian.

A detailed discussion is beyond the scope of this experimental work, but can be found in the new theory manuscript cited above. To distinguish this from a true stationary state (or equilibrium) of the system, such as a thermal state fulfilling the condition (2), we consistently refer to it as the "asymptotic steady state" or "infinite time average" and precisely define what we mean in the Methods. The effective equilibration is an intrinsic feature that has nothing to do with thermalization. Nevertheless, the equilibrium energies that a slowly thermalizing refrigerator would eventually assume would be close to the values given by the asymptotic steady state of the unitary model.

For this type of closed system I would expect that the observables, instead of evolving to a proper limit, would evolve in a quasi-periodic manner. However, it may very well be the case that the relevant observables may stay close to an "equilibrium value" for long stretches of time, although deviating from it occasionally. I get the impression that the authors are aware of this, due to the paragraph in the Methods-section on page 12 where they cite [28] and [29], where these notions are discussed. Thus, an alternative way to phrase the results of this study would be to say that it compares the performance of this isolated thermal machine, for the case when the evolution time is selected specifically to optimize the performance, with the case when we pick "generic" long evolution times, where the relevant observable is close to its equilibrium value, in the sense of closed system equilibration. I would thus suggest that the authors consider the possibility to, instead of phrasing their study as a direct quantum implementation of the standard notion of absorption refrigerators, phrase it as an analogous construction, but in a closed system scenario, and explicitly discuss and highlight both differences and similarities between the two. (I would also avoid to use the terms "steady state" and "long term limits" in the closed system scenario.) I also think it could be fruitful to explicitly discuss (and potentially investigate a bit further) to what extent one can understand the behavior of this device in terms of closed system equilibration. I might add that this topic has witnessed a rather vigorous development, and a clear connection could potentially boost a wider interest for this study.

We agree with the referee that the connection to the topic of closed system equilibration is remarkable and calls for further studies. The above mentioned theory manuscript sheds more light on this somewhat surprising feature of the trilinear interaction Hamiltonian. However, we would like to stress that the closed system evolution of an initially uncorrelated thermal three-mode state is

not even quasi-periodic, because of the incommensurate energy spectrum of the Hamiltonian. As mentioned before, our simulations showed that no recurrence of the initial energies takes place even after very long stretches of interaction time. (We did a simulation run up to $t = 10000/\xi$ to be sure; the exemplary plot in the appendix of the theory manuscript reaches $200/\xi$. The residual oscillation amplitudes at long times decay rapidly with the initial temperatures, i.e. the populated effective dimension of the problem.) We think that the equilibration is a truly remarkable feature, given the comparable simplicity of the interaction term and the fact that its classical counterpart is integrable, see the theory manuscript. In the present work, we demonstrate this feature, but refrain from discussing the theoretical background in detail.

Yet to explain effective equilibration in a closed system, we need to introduce the notions of an infinite time average or asymptotic steady state, the precise definition of which we give in the Methods section. We do not try to suggest that the unitarily evolving state ever converges towards a steady state, but rather that the mode energies quickly approach and stay close to the values associated to the infinite time averaged state. This state is identical to the fully dephased counterpart of the initial state and is thus a steady state under the trilinear Hamiltonian. (In "Numerical simulation" of the Methods section, we discuss a phase-incoherent implementation of the trilinear energy exchange that leads to this steady state.)

The mentioned theory manuscript also makes an exemplary comparison between the unitary closed-system evolution of the modes and an open-system evolution under weak coupling to local thermal reservoirs (with thermalization rates to the hot, cold, and work reservoir much lower than the coupling rate ξ). The asymptotic long-time averages differ slightly, but the transient dynamics of both models match closely. Therefore we state in the Methods that the unitary model studied here approximates "the fast internal dynamics of an absorption refrigerator whose thermalization rate with the reservoirs associated to each mode is much slower."

All the above comments concerning "steady states" and "long term limits" relate to the closed system scenario.

However, another possibility is that there actually is some additional irreversible influence from the environment. For example, in addition to the Hamiltonian evolution there could be some degree of decoherence with respect to the eigenbasis of the global Hamiltonian (for example as described by the master equation on page 17, although one could of course combine the Hamiltonian evolution with a dissipator of this type). This would yield an irreversible approach to the decohered state in the energy eigenbasis, and a well defined limit state. (Limit states may also exist for other open system effects that do yield decohere to the energy eigenbasis.)

If the experimental conditions are such that there is a significant degree of decoherence, or not, is beyond me to judge. Unfortunately, the only comment that I can find in the manuscript that touches this issue is on page 13 in the Methods-section, where it is stated that the coherence time of a single phonon is much larger than the time required to achieve the "steady state". The question whether the evolution in this experiment can be regarded as purely Hamiltonian to a good approximation, or if there are some significant open system effects, appears to me to be rather fundamental for this study, and I think that it deserves more attention and should be discussed more thoroughly. Moreover, if it indeed would be obvious that the evolution is Hamiltonian, and described by (1), then one might wonder why the experiments were done in the first place, as it then seems like one equally well might have resorted to purely numerical investigations.

I cannot judge the feasibility, but maybe the authors could consider the possibility to complement their study with some more direct probing of the quality of the Hamiltonian description for this system?

There are several mechanisms that may lead to decoherence in our system, however all of them happen at much longer time scales than $1/\xi$.

First of all, there is heating of the motional modes due to the electric field noise from the trap electrodes. For the modes of interest this heating increases the energy of the mode by about 2 phonons / second.

Secondly, there is a dephasing of the motional modes which we attribute to the fast fluctuations of the trap frequency. It leads to decoherence of the superposition of Fock states $(|0\rangle + |1\rangle)/\sqrt{2}$. The coherence time of the modes measured in our setup is greater than 8 ms for all the modes.

Finally we also checked (for slightly different trap frequencies than used in the rest of the paper) the decay time of the coherent energy exchange between the modes under the trilinear hamiltonian evolution, if one starts with a single phonon Fock state in hot mode and vacuum states in cold and work modes. We observe substantial reduction of the oscillation amplitude only after about 200 oscillations.

All these clearly shows that for the relevant time scales of the experiment the coupling to the environment is weak and the evolution of the system can be treated as unitary.

We have expanded the discussion of the possible decoherence mechanisms in the Method section of our paper as outlined above.

Moreover, although the experiments agree better with the results derived from (1) than those obtained from (2), there seems to me to still be rather significant deviations. Do the experiments give any hint of how one might improve the model beyond (1)? Or do the authors have some intuition concerning this? In general I believe that experimental hints on how we could improve the model could give an added value to this study.

We believe that most of the observed deviations can be attributed to experimental imperfections, rather than to the model failures (see the detailed reply to referee 1 and updates to the manuscripts on the error analysis). The model predicts the coherent evolution of the modes to some asymptotic steady state and we verify that no noise sources influence this evolution on the relevant time scale (see reply above). However, the overall measurement time required for obtaining reasonable statistics for the experimental points is quite long and several instrumental errors might accumulate due to slow drifts of the experimental parameters. We believe that improving the stability of the apparatus will allow to eliminate the uncertainties and the model (1) should be better reproduced in the experiment.

As a very minor remark, in equation (2) I presume that the parenthesis on the left hand side of the equality sign does not play any role and could be removed.

We have removed the parenthesis in equation 2.

Reviewer #3 (Remarks to the Author):

This paper discusses the experimental realization of a quantum absorption refrigerator with 3 trapped ions. The paper aims to compare the cooling capability when using thermal states or squeezed states, and the performance in the single shot regime with the steady state limit.

The results show that cooling is most effective when there is no squeezing at all. It also shows that when the interaction is switched off at the right moment, so that the system does not reach the steady state, the absorption fridge cools quicker and stronger than in the steady state case.

This is an exciting experiment that will be valuable for the quantum thermodynamics community in testing one of the much discussed heat machines, the quantum absorption refrigerator, for the first time experimentally.

I found the paper clear in the story it is explaining, but had significant problems understanding how the theoretical concepts are realized in the experiment and what the experimental data shown in the figures indicate.

Therefore I recommend that the manuscript is potentially suitable to be published in Nat Comm, but the authors need to revise the manuscript substantially to make it accessible and scientifically sound.

Specific queries are the following:

Eq. (1): the Coulomb interaction is a long-range binary interaction. Further details of how the trip-linear interaction comes about and how bi-linear interactions are avoided in the experiment are required.

We added a Supplementary Material section with a detailed derivation of the trilinear interaction Hamiltonian starting from the Coulomb interaction between the modes. We also explain there that the nonlinear coupling between pair of modes is avoided since the resonance conditions to induce such coupling are not satisfied. Bi-linear interactions between harmonic modes of motion do not arise in our system, because the normal modes of motion that we choose to describe the system are independent, and do not interact with each other in the absence of anharmonicity.

Also a clearer exposure in the main text explaining what constitutes the 3 ion modes, that make up the absorption refrigerator, is needed. (it's in the methods, but should be included in the main text)

We moved the definition of the normal modes used in the experiment to the main text of the paper (caption of Figure 1). We display the modes on Fig 1 and label them in the caption.

Fig 2: comparison of data with Eq.(2) shows deviations, as mentioned in the text.

But why should (2) hold in the first place and what is the physical reason it doesn't actually?

Or alternatively, maybe (2) shouldn't hold, why not?, and then why is there a point comparing the experimental data with it?

As we also mentioned in the reply to the referee 1, given an initial state with temperatures out of equilibrium, the formula gives the thermal equilibrium temperatures that the modes could, in principle, reach if the exchange of energies between them is governed by the Hamiltonian (1) but the modes remain in the thermal states (which would be the case if the heat bodies were classical). We do expect equation (2) to hold if the initial temperatures of the modes satisfy equilibrium condition (Fig. 2e). However, we don't expect that the asymptotic steady state the coupled system approaches is a product of the thermal states if we start away from equilibrium (as shown in Fig 2a-2d). Nevertheless, we prefer to compare our data with equations (2) and (3) everywhere, since they provide a useful comparison with the equivalent classical system and highlight the fact that the asymptotic steady state is not simply a product of thermal states if the system was prepared away from equilibrium.

We expanded the paragraph following equation (2) in order to clarify this.

page 5: Could the authors add a sentence explaining why $E_c = E_w = E_h$?

The structure of the interaction Hamiltonian (1) is such that removing a quantum from the hot mode implies adding a quantum in the work and in the cold mode, and vice versa. Consequently, any change in the mean phonon number in the hot mode over time must imply the same opposite change in the work and in the cold mode.

page 7: it says "we next investigate ... quantum mechanical coherence ... we prepare a squeezed thermal state": what are the coherences of this state? The subsequent discussion mentions that the mean photon

number changes as a result of squeezing. So isn't that the main effect of squeezing/in what sense is there a link to coherence?

We have addressed a similar question regarding the motivation to use the coherent state in the reply to the Referee 1. We further add here that the squeezing results in not only the increase of the phonon numbers, but also in the appearance of off-diagonal elements (coherences) of the density matrix of this state written in Fock basis, and in reduction of the quadratures of the squeezed state.

As we already mentioned in the reply to referee 1, there is an ongoing discussion in the field of quantum thermodynamics regarding the nonequilibrium thermal bath and its effects on the heat machine performance, and several theoretical studies claim that a squeezed thermal bath is able to enhance the performance of the quantum heat machines.

We don't want to pick sides in the discussion and prefer to present our data and leave it open to interpretation by the reader.

Fig 4. contains an incomplete sentence referring to (2).

We have rewritten the sentence.

the word "nett" probably should be "net" throughout

The word "nett" is a spelling variant of "net", commonly used in UK and Singapore. Nature family journals follow British spelling.

Methods:

page 14: Why do the authors chose to first cool (sideband) and then heat again?

It is done to achieve reproducible preparation of the initial states. Temperature of the initial states after doppler cooling alone is not very well controlled in the experiment and depends strongly on the small changes of the laser frequencies and powers. We added a phrase: "To achieve reproducible preparation of the initial states ... " to the section explaining the preparation of the initial states.

(I like the sweat drops, sunglasses and hats of the work, warm and cold modes, respectively, in Fig 1.)

We thank referee for this remark, we left pictures of ions on Fig 1 unchanged.

Reviewers' comments:

Reviewer #1 (Remarks to the Author):

Dear editor,

in the revised version of their manuscript, the authors have successfully adapted the data analysis, which was my main concern in the previous report. Furthermore, the concept of "unitary refrigeration" is now linked to effective equilibration due to incommensurate frequencies, which renders it viable and puts it into context with "quantum thermodynamics". Furthermore, parts of the main text were rewritten to make the manuscript more accessible. In the previous report, I already stated that the general idea of demonstrating refrigeration in a well-controlled few-body system via nonlinear conversion is quite compelling, therefore I now suggest acceptance of the manuscript for publication in Nature Communications.

I would however still ask the authors to address a couple of points:

1. Data presentation in Fig. 2: This figure should demonstrate the general refrigeration effect. Both upon the first read and upon rereading the revised version, it took me long time to understand figure and how to infer that the refrigeration actually work. In Figs. 3 & 4, the cold mode phonon number difference is plotted versus the work mode phonon number, which allows for direct interpretation. I would suggest to simplify Fig. 2 and improve the explanation, e.g. add a few sentences on why the data is presented in this way.
2. Spectator modes: In the previous report, I expressed the concern that spectator modes are not taken into account in the data evaluation. The authors now include off-resonant coupling to spectator modes during the probing, which leads to rather small errors. However, large errors can be introduced for strongly excited spectator modes due to dispersion of the respective carrier Rabi frequencies. Taking spectator modes into account, summation over all modes with nonzero coupling have to be taken into account in equation S2, and the total Rabi frequency contains the carrier Rabi frequencies of the spectator modes. I do NOT ask the authors to redo the entire evaluation taking this into account, but adding a corresponding statement and estimating the magnitude of the errors would complete the thorough data analysis.
3. Quantum effects: The title and introduction of the manuscript strongly suggest that quantum mechanics fundamentally underlies the working principle of the refrigerator. I am not entirely convinced that this is true. The question to be asked would be if a classical molecular dynamics simulation of the system would yield different results? As the working principle relies on nonlinearity / incommensurate mode frequencies, my intuition would be that this could indeed be the case. In line 19, the authors state "We also study the performance of such a refrigerator ... made possible by quantum coherence". What exactly would "quantum coherence" be in this case? A more clearly defined notion is the one of a "non-classical state", however the states of clearly non-classical nature occurring in this work –the squeezed states- do not lead to a performance improvement. I would ask the authors to reconsider the strong "quantum" selling point of this work.
4. Figure 3: I expressed concerns about the validity of the fits in panel a-f and h-k. While the authors correctly state that "68% of the experimental points should be within 1 standard deviation from the theoretical simulations", which approximately holds, an analysis e.g. in term of R^2 would yield a bad descriptive power of the fits. As the important data are the steady-state values, which are correctly described, one could consider not presenting these plots, as it is unclear anyway how these contribute to the general understanding.
5. Other work on nonlinearity with trapped ions: I would also ask the authors to consider citing work on nonlinearities with ion crystals from other groups, e.g. "Nonlinear coupling of continuous variables at the single quantum level", Phys. Rev. A 77, 040302(R) (2008)

In summary, this is outstanding and interesting work, and I do not ask the authors to make significant changes in the final revision, I am merely asking to adjust the presentation such that appearance of the manuscript better matches its content.

Reviewer #2 (Remarks to the Author):

I am satisfied with the modifications that the authors have done in response to the concerns that I raised in my previous report. The material and findings are now much better presented in relation to previous notions. In particular, the accompanying theoretical paper [27] is indeed very helpful for clarifying the general ideas.

I have no further comments on the manuscript, and in view of the so far limited experimental studies on quantum thermal machines, and since I believe that this approach could inspire future experiments and theoretical studies, I would say that publication in Nature Communications is justified.

Reviewer #3 (Remarks to the Author):

The authors have provided a substantial reply to the concerns raised by 3 referees. They have also redone the data analysis and changed the manuscript.

Unfortunately my assessment of the paper has not changed.

On the positive side, this is an exciting experiment that will be valuable for the quantum thermodynamics community in testing one of the much discussed heat machines, the quantum absorption refrigerator, for the first time experimentally.

On the negative side, the paper does not communicate clearly what is being done.

For example, the Results section starting with "To demonstrate ..." doesn't sufficiently explain what the preparation is, how the system is controlled, how it is driven, how it is measured, and what different setups are being considered. The three modes used are now introduced in the paper (great!), but only in the caption of Figure 1 (why?) without further text describing the setup and how it realises an absorption fridge. The following text describes plots in Fig 2 and further discusses the findings, but without having prepared a clear ground to build on.

I'm afraid the paper is not clear to a generalist reader that should be assumed for Nature Communications.

While I like many aspects of the scientific content of the paper, the presentation is not suitable for publication.

Reply to the Reviewers' comments

We want to thank all Reviewers for the second assessment of our manuscript. We are glad that our previous reply and manuscript updates have successfully addressed much of the criticism and clarified the content. In the second round of response to Reviewer comments the greatest changes to the manuscript comes from the attempt to reply to Referee 3 who remains sceptical about the clarity of the paper to generalist reader. In response to that, we have substantially revised the Introduction part of the manuscript. In particular, we have added more discussion on the technical realization of the absorption refrigeration in the trapped ion system, clarified the labelling of the refrigerator bodies in terms of normal modes and moved the expanded theoretical explanations into the Results section. In response to Referee 1 comments, we have update Figure 2 to achieve better readability, re-assessed the contribution of spectator modes to the measured phonon numbers and updated the manuscript to achieve more clarity of “quantum” versus “classical” nomenclature of the underlying processes. We hope that current version of the the manuscript is acceptable for publication in Nature Communications.

Below we give detailed response to the referee's comments and suggestions. As before, our response is typed in blue, and the original referee comments in black.

Reviewer #1 (Remarks to the Author):

In the revised version of their manuscript, the authors have successfully adapted the data analysis, which was my main concern in the previous report. Furthermore, the concept of “unitary refrigeration” is now linked to effective equilibration due to incommensurate frequencies, which renders it viable and puts it into context with “quantum thermodynamics”. Furthermore, parts of the main text were rewritten to make the manuscript more accessible. In the previous report, I already stated that the general idea of demonstrating refrigeration in a well-controlled few-body system via nonlinear conversion is quite compelling, therefore I now suggest acceptance of the manuscript for publication in Nature Communications.

I would however still ask the authors to address a couple of points:

1. Data presentation in Fig. 2: This figure should demonstrate the general refrigeration effect. Both upon the first read and upon rereading the revised version, it took me long time to understand figure and how to infer that the refrigeration actually work. In Figs. 3 & 4, the cold mode phonon number difference is plotted versus the work mode phonon number, which allows for direct interpretation. I would suggest to simplify Fig. 2 and improve the explanation, e.g. add a few sentences on why the data is presented in this way.

We followed the referee suggestion and updated Fig. 2e to clearly show the region of parameters where the refrigeration condition ($T_c < T_h < T_w$) is met. We have omitted repeating labels in panels a-f to improve readability. The text in the Results section was expanded to provide more insight on how the points in Fig 2e were obtained and also how to draw a comparison between panels a-d of this Figure and the data presented in Figs 3 and 4.

2. Spectator modes: In the previous report, I expressed the concern that spectator modes are not taken into account in the data evaluation. The authors now include off-resonant coupling to spectator modes during the probing, which leads to rather small errors. However, large errors can be introduced for strongly excited spectator modes due to dispersion of the respective carrier Rabi frequencies. Taking spectator modes into account, summation over all modes with nonzero coupling have to be taken into account in equation S2, and the total Rabi frequency contains the carrier Rabi frequencies of the spectator modes. I do NOT ask the authors to redo the entire evaluation taking this into account, but adding a corresponding statement and estimating the magnitude of the errors would complete the thorough data analysis.

In his comment, the referee points out equation S2 as the one to be modified in order to consider the effects of carrier Rabi frequency dispersion. We want to note, that equation S2 was used to fit the temporal scans of the blue sideband evolution during calibration of the motional state preparation. During those scans only one mode of interest was controllably heated away from the ground state and the carrier contribution (if any) was taken into account as the background parameter during fitting procedure. The other modes remained close to the ground state and their contribution to the carrier Rabi frequency should have remained constant.

However, as the referee correctly points out, during the measurement procedure, we are dealing with several populated modes whose population can have an effect on the carrier Rabi frequency and therefore alter the measured brightness due to off-resonant coupling of the carrier to the detected mode. We illustrate and estimate this effect for the lowest frequency of the cold mode used in the experiment (see Figure S1 b). The contribution of the carrier at the frequency of this mode can be influenced by a changing population in radial rocking work mode only. We note that all other modes shown in Figure S1 b remain close to the ground state during the whole experimental sequence.

During motional state detection we are probing the red sideband of the cold mode as described by equation 7 with a pulse duration of $t_{rsb} = \frac{\pi}{3\Omega}$, where Ω is the Rabi frequency of the motional sideband after sideband cooling, which is detuned from the carrier transition by Δ .

To estimate the contribution of the carrier to the detected signal we have assumed that the modes remain in a thermal state and modified equation 7 to include thermal distributions of the work and cold mode as two independent sources that affect the carrier Rabi frequency.

The carrier Rabi frequency for the cold mode in the Fock state n and work mode in the Fock state m is:

$$\Omega_c = \Omega_0 (e^{-\frac{\eta_{cold}^2}{2}} L_n(\eta_{cold}^2)) (e^{-\frac{\eta_{work}^2}{2}} L_m(\eta_{work}^2)) ,$$

where Ω_0 is the carrier Rabi frequency, η_{cold} (η_{work}) is the Lamb-Dicke parameter for the cold (work mode) and L is the Laguerre polynomial. We have evaluated the carrier Rabi

frequency $\Omega_0 = \frac{1}{2}(\frac{\Omega_{cold}}{\eta_{cold}} + \frac{\Omega_{work}}{\eta_{work}})$ from the experimentally measured Rabi frequencies for cold and work blue motional sidebands after sideband cooling (see Supplementary Table S2). Finally, the probability to find the ion in the $|\uparrow\rangle$ state at detuning Δ from the carrier is:

$$p_{\uparrow}(\bar{n}, \bar{m}, \Delta) = \sum_{n=0}^{\infty} \sum_{m=0}^{\infty} \frac{\bar{n}^n}{(\bar{n}+1)^{n+1}} \frac{\bar{m}^m}{(\bar{m}+1)^{m+1}} \frac{\Omega_c^2}{\Omega_c^2 + \Delta^2} (\sin^2(\sqrt{\Omega_c^2 + \Delta^2} \frac{t_{rsb}}{2}))$$

The spectrum of the radial mode carrier and red sidebands of the cold and work modes is shown below. The carrier peak is plotted for several values of mean phonon numbers used in the paper and the change in its shape may induce changes in measured brightness at cold mode frequency.

The Figure below shows the carrier induced error in the measured cold mode mean phonon number with $\bar{n} \approx 2.7$ while work mode mean phonon number varies from 0 to 10. We see that a systematic error due to the dispersion of the carrier Rabi frequencies contributes to less than a percent of the measured mean phonon number. It is almost negligible when compared to the statistical uncertainty of our measurement.

We have estimated the error contribution for the worst case scenario using largest value of the gradient $\frac{\partial n_{th}}{\partial p_1}$ that occurs in our system. The smaller values of the gradient as well as the evaluation of these errors for larger detection mode detuning would lead only to the decrease of the error. We hope that this evaluation completes the error analysis.

We have added a following sentence to the end of the Methods section:

“Also, equation 7 does not include off-resonant contributions of the carrier transition which Rabi frequency depends on the population of the excited modes. However, we have estimated that this effect gives systematic errors of less than 1% to the reconstructed mean phonon numbers.”

3. Quantum effects: The title and introduction of the manuscript strongly suggest that quantum mechanics fundamentally underlies the working principle of the refrigerator. I am not entirely convinced that this is true. The question to be asked would be if a classical molecular dynamics simulation of the system would yield different results? As the working principle relies on nonlinearity / incommensurate mode frequencies, my intuition would be that this could indeed be the case. In line 19, the authors state “We also study the performance of such a refrigerator ... made possible by quantum coherence”. What exactly would “quantum coherence” be in this case? A more clearly defined notion is the one of a “non-classical state”, however the states of clearly non-classical nature occurring in this work –the squeezed states- do not lead to a performance improvement. I would ask the authors to reconsider the strong “quantum” selling point of this work.

We thank the referee for pointing out the issue of quantum-classical comparison. This is indeed an interesting point that some of the authors have addressed in a theory publication,

see Ref. 26. The refrigeration features observed in this work, including the single-shot cooling related to inter-mode coherence, could be explained qualitatively in a purely classical framework, and we state this now explicitly in the paragraph before Eq. (1). Moreover, we removed the 'quantum' from 'quantum coherence' in the abstract, and we now introduce the idealized benchmark in Eq. (3) without using the imprecise reference to a 'classical description'.

Nevertheless, we point out that the correct quantitative prediction of the fridge dynamics we observe here requires a quantum description of the ions, as we are dealing with motion that is excited not far from its ground state. So while there is indeed no genuine advantage in performance associated to the quantum nature of the ions' dynamics and thermal statistics, we are still clearly demonstrating absorption refrigeration in the quantum regime.

4. Figure 3: I expressed concerns about the validity of the fits in panel a-f and h-k. While the authors correctly state that "68% of the experimental points should be within 1 standard deviation from the theoretical simulations", which approximately holds, an analysis e.g. in term of R^2 would yield a bad descriptive power of the fits. As the important data are the steady-state values, which are correctly described, one could consider not presenting these plots, as it is unclear anyway how these contribute to the general understanding.

We would like to point out that the solid lines in panels a-f and h-k of Fig. 3 are not fits to the experimental data. They are numerical simulations of the cold mode evolution for given initial conditions. We feel that showing the overlap of the experimental points with theoretical predictions that do not include any free parameters proves that we can control the system fairly well during the evolution. These panels serve as a good guideline for observation of a transition from cooling to heating in the cold mode through the equilibrium state (panel c) and is necessary for understanding the single-shot cooling mechanism, since the clear minimum in the cold mode energy can be observed in panel a. We would like to keep the panels in the manuscript.

5. Other work on nonlinearity with trapped ions: I would also ask the authors to consider citing work on nonlinearities with ion crystals from other groups, e.g. "Nonlinear coupling of continuous variables at the single quantum level", Phys. Rev. A 77, 040302(R) (2008)

We have added the suggested reference.

In summary, this is outstanding and interesting work, and I do not ask the authors to make significant changes in the final revision, I am merely asking to adjust the presentation such that appearance of the manuscript better matches its content.

We would like to thank the referee for the comprehensive review of our manuscript.

Reviewer #2 (Remarks to the Author):

I am satisfied with the modifications that the authors have done in response to the concerns that I raised in my previous report. The material and findings are now much better presented

in relation to previous notions. In particular, the accompanying theoretical paper [27] is indeed very helpful for clarifying the general ideas.

I have no further comments on the manuscript, and in view of the so far limited experimental studies on quantum thermal machines, and since I believe that this approach could inspire future experiments and theoretical studies, I would say that publication in Nature Communications is justified.

Reviewer #3 (Remarks to the Author):

The authors have provided a substantial reply to the concerns raised by 3 referees. They have also redone the data analysis and changed the manuscript.

Unfortunately my assessment of the paper has not changed.

On the positive side, this is an exciting experiment that will be valuable for the quantum thermodynamics community in testing one of the much discussed heat machines, the quantum absorption refrigerator, for the first time experimentally.

On the negative side, the paper does not communicate clearly what is being done.

For example, the Results section starting with “To demonstrate ... “ doesn't sufficiently explain what the preparation is, how the system is controlled, how it is driven, how it is measured, and what different setups are being considered. The three modes used are now introduced in the paper (great!), but only in the caption of Figure 1 (why?) without further text describing the setup and how it realises an absorption fridge. The following text describes plots in Fig 2 and further discusses the findings, but without having prepared a clear ground to build on.

I'm afraid the paper is not clear to a generalist reader that should be assumed for Nature Communications.

While I like many aspects of the scientific content of the paper, the presentation is not suitable for publication.

We went through the manuscript again and added several sentences that introduce the mode setup and coupling dynamics explicitly. This includes a passage in the paragraph before Eq. (1) that now names the motional modes, rewriting the text before Eq. (2) that now explains the experimental procedure together with the working principle of the refrigerator, and a rephrasing of the 1st paragraph in the Results section mentioned by the referee. We hope that this gives a clearer picture for the general reader.

REVIEWERS' COMMENTS:

Reviewer #1 (Remarks to the Author):

The authors have successfully addressed all remaining points. The presentation now provides sufficient clarity, and i have confidence that the data analysis is valid. I therefore have no further objections agains publication of the manuscript in Nature Communications.

Reviewer #3 (Remarks to the Author):

The authors have made the requested additions in the introduction/results to clarify the modes and working principle of the fridge.

The paper is now in good shape and, as said before, reports on a very nice and important experiment.

I am happy to recommend acceptance for Nature Comms.

Minor comments for consideration by the authors:

line 100 - do you mean Eq (3)?

Fig 3 caption: "purely thermal a-f" -> "purely thermal state (a-f)"

and "for thermal g" -> "for thermal state (g)"

Reply to the Reviewers' comments

Once again we would like to thank all Reviewers who helped us to improve our manuscript. We are happy that our previous replies and manuscript updates cleared up the remaining concerns of the Reviewers. In the current version of the manuscript we corrected the typos noticed by Reviewer 3. We have also formatted the manuscript to comply with the editorial requirements. We hope that this version of the the manuscript is acceptable for publication in the Nature Communications.

Below is our response to the Reviewer 1 and 3 comments. As before, our response is typed in blue, and the original referee comments in black.

Reviewer #1 (Remarks to the Author):

The authors have successfully addressed all remaining points. The presentation now provides sufficient clarity, and i have confidence that the data analysis is valid. I therefore have no further objections agains publication of the manuscript in Nature Communications.

We thank the Reviewer for the comprehensive review.

Reviewer #3 (Remarks to the Author):

The authors have made the requested additions in the introduction/results to clarify the modes and working principle of the fridge.

The paper is now in good shape and, as said before, reports on a very nice and important experiment.

I am happy to recommend acceptance for Nature Comms.

Minor comments for consideration by the authors:

line 100 - do you mean Eq (3)?

Fig 3 caption: "purely thermal a-f" -> "purely thermal state (a-f)"
and "for thermal g" -> "for thermal state (g)"

We thank the Reviewer for the detailed review. We have corrected the typos in the manuscript.